# Glial Plasticity and Metabolic Stability After Knockdown of Astrocytic Cx43 in the Dorsal Vagal Complex

**DOI:** 10.3390/cells14211694

**Published:** 2025-10-29

**Authors:** Manon Barbot, Bruno Lebrun, Rym Barbouche, Stéphanie Gaigé, Alain Tonetto, Anne Abysique, Jean-Denis Troadec

**Affiliations:** 1Centre de Recherche en Psychologie et Neurosciences (CRPN), CNRS UMR 7077, Aix-Marseille University, 13331 Marseille, France; manon.barbot@outlook.com (M.B.); bruno.lebrun@univ-amu.fr (B.L.); rym.barbouche@univ-amu.fr (R.B.); stephanie.rami@univ-amu.fr (S.G.); anne.abysique@univ-amu.fr (A.A.); 2Plateforme de Recherche Analytique Technologique et Imagerie (PRATIM), FSCM (UAR1739), CNRS, Centrale Marseille, Aix-Marseille University, 13331 Marseille, France; alain.tonetto@univ-amu.fr

**Keywords:** brainstem, astrocytes, microglia, gap junctions, hemichannels, food intake, obesity, plasticity

## Abstract

**Highlights:**

**What are the main findings?**

**What is the implication of the main finding?**

**Abstract:**

Obesity causes millions of deaths each year due to metabolic complications, making it a major public health challenge. It results from a chronic imbalance between caloric intake and energy expenditure. Among central structures regulating energy balance, the dorsal vagal complex (DVC) integrates metabolic signals from energy stores and the gastrointestinal tract and coordinates autonomic responses. While historically overshadowed by a focus on neurons, the role of glial cells in regulating energy balance is now well established. Connexin 43 (Cx43) is a well-known protein expressed by astrocytes, playing a key role in glial and neuroglial communication. To investigate the role of astrocytic Cx43 within the DVC, where its expression is remarkably high, we specifically reduced it using an RNA interference approach. Although reduced Cx43 expression led to modified astrocyte and microglia morphology and phenotype, our analyses did not reveal significant changes in the animal’s metabolic phenotype under standard feeding conditions as well as under a high-fat, high-sugar diet. These results suggest that denser astrocytic tiling and hyper-ramified microglia may constitute a buffering system that preserves metabolic and autonomic outputs when a single connexin pathway fails.

## 1. Introduction

The DVC is a brainstem structure that plays a central and multifaceted role in the control of energy balance. By integrating gastrointestinal sensory afferent information, the DVC coordinates efferent responses to regulate food and caloric intake, glucose homeostasis, pancreatic exocrine secretion, and gastric/intestinal motility and emptying. Most of these gastrointestinal sensory inputs are transmitted by the vagal afferents projecting to the nucleus of the solitary tract (NTS) and, to a lesser extent, to the area postrema (AP). In addition to vagal afferents, NTS neurons receive hormonal and metabolic circulating information related to adiposity, glycemia, and nutrient availability through their proximity to the AP, a circumventricular organ with fenestrated capillaries and a leaky blood–brain barrier (BBB) [1,2]. Recently, several studies have reported that in addition to neuronal networks (see [3,4,5,6] for review), glial cells contribute to the regulation of feeding behavior and energy balance (see [7,8] for review). Astrocytes in the DVC detect metabolic changes and respond to hormones signaling metabolic status, such as leptin [9,10] or GLP-1 [11], thereby influencing synaptic transmission [12,13]. Astrocytes undergo morphological changes in response to a high-fat diet (HFD), which coincide with behavioral modifications. Notably, there is a transient early phase during the first week of HFD exposure that is associated with significant hyperphagia [13,14,15], followed by a later reappearance of these changes in obese animals that have consumed an HFD for several weeks [9]. In 2020, MacDonald and colleagues [15] reported that the selective activation of astrocytes using a DREADD-based approach reduced food intake and slowed weight regain after a 12-h fast. This activation increased local neuronal activity, as evidenced by increased c-Fos expression [15]. In this context, to better characterize the role of glial communication into the DVC, we focused on Cx43, a key protein involved in the formation of gap junctions between astrocytes. In the CNS, glia, especially astrocytes, express different connexins [16] (Cxs;). These Cxs constitute gap junctions (GJs) that contribute to cytoplasmic continuity, providing the structural basis for an extensive astroglial network [17]. Alternatively, Cxs not engaged in GJ form hemichannels (HCs), which are permeable to small molecules, such as glucose, ATP, D-serine, and glutamate [18]. HCs and GJs work in synergy to coordinate, via autocrine or paracrine signaling, the activity of glial cells and neurons, in response to environmental cues [19]. Moreover, Cx43-mediated communication allows astrocytes to functionally link neuronal networks that are not directly connected via synapses, by releasing gliotransmitters that modulate synaptic transmission and neuronal excitability [20,21]. We recently reported that Cx43 is highly expressed by DVC glial cells and often found near synaptic elements [22]. Allard et al. [23] provided the first indications of astrocytic communication via Cx43 in hypothalamic glucose sensing. By administering siRNAs targeting Cx43 in the hypothalamic parenchyma of rats, they transiently inhibited Cx43 expression. This inhibition led to reduced insulin secretion in response to intracarotid glucose injection. Furthermore, lactate trafficking through Cx43 gap junctions in the network of hypothalamic tanycytes is required to maintain POMC neuronal activity and selective deletion of tanycytic Cx43 was shown to alter feeding behavior and energy metabolism [24]. More recently, intracellular ATP has also been reported to be released via tanycytic Cx43 HCs and extracellularly converted into adenosine by ectonucleotidases which in turn downregulate Agouti Related Protein (Agrp) and neuropeptide Y (NPY) expression and exerted an anorexigenic effect [25]. Moreover, we recently reported that intracerebroventricular (ICV) injection of TAT-GAP19, a specific blocker of Cx43 HCs, decreased food intake and induced neuronal activation limited to the hypothalamus and DVC [22]. In the present work, to investigate the contribution of Cx43 to the brainstem regulation of energy homeostasis, we developed an approach using adeno-associated viral vectors (AAV/DJ8) under the control of glial fibrillary acidic protein (GFAP to express shRNAs directed against Cx43 specifically in DVC astrocytes. After verifying the specificity and efficacy of Cx43 deletion in DVC astrocytes, we studied the organization of astrocytes and microglial networks and characterized the metabolic phenotype of these animals under both standard AO4 (NC) and high-fat/high sugar (HFS) diets.

## 2. Materials and Methods

### 2.1. Animal, Diets, and Longitudinal Monitoring

Adult male C57BL/6J mice were obtained from Charles River Laboratories (L’Abresle, France). They were housed in standard cages under controlled environmental conditions, including a 12-h light-dark cycle (7:00 a.m. to 7:00 p.m.) and an ambient temperature between 25 °C and 28 °C. Before AAV injection surgery, mice were housed in groups of 3 or 4. After surgery, they were housed individually for the remainder of the study. They had free access to water and food, except during protocols that required fasting. Mice were fed either NC pellets (SAFE, Augy, France) or HFS diet (Sniff #E15744-34, Soest, Germany). The HFS diet was introduced at 5 weeks of age, 3 weeks before the surgical procedure for AAV injections. All mice in this study were weighed at least twice a week. Between weeks 5 and 9 post-injection, the mice underwent various tests to assess energy metabolism. During periods without specific tests, food intake was monitored in some cohorts, measured using the leftover weighing method. Weekly blood glucose measurements were performed on some cohorts. Blood samples were collected from the tail tip and analyzed directly using test strips and a glucometer (Accu-Chek Performa Nano, Roche Diagnostics Corporation, Indianapolis, IN, USA).

### 2.2. Adeno-Associated Viral Vectors (AAV) and Stereotaxic Delivery

Two distinct adeno-associated viral vectors (AAV/DJ8) were designed to express the fluorescent protein EGFP under the control of the GFAP promoter. One vector, AAV-DJ/8-GFAP::EGFP/shRNA-Scramble contained an inactive shRNA, while the other vector, AAV-DJ/8-GFAP::EGFP/shRNA-Cx43, contained three shRNAs targeting Cx43 (VectorBuilder, Neu-Isenburg, Germany; See Appendix A for the plasmids used for generating the AAVs). These vectors are referred to as AAV-shRNA-Scramble and AAV-shRNA-Cx43. Each vector solution was prepared at a concentration of 2 × 10^11^ genome copies/mL in sterile 0.9% NaCl solution and was stereotactically injected bilaterally into the DVC of 8-week-old mice.

The protocol for injections into the DVC parenchyma was developed based on the work of Brierley et al. [26], using a trans-atlanto-occipital approach. Before surgery, mice received intraperitoneal injection of analgesia (buprenorphine, 0.05 mg/kg, Ceva Santé Animale, Libourne, France) and anesthesia with a mixture of ketamine (110 mg/kg, Imalgène 1000, Boehringer Ingelheim, Lyon, France) and xylazine (12 mg/kg, Rompun 2%, Bayer Santé, Lyon, France). The skin on the back of the head and neck was shaved, and local anesthesia was administered to the scalp (lidocaine, 2 mg/kg, Vetoquinol, Lure, France). The mice were positioned in a stereotaxic frame, with their heads angled at 90° relative to their necks. Under aseptic conditions, a 5 mm incision was made along the skin, and the neck muscles were gently retracted to expose the atlanto-occipital membrane and the obex, which served as an anatomical reference. The membrane was incised horizontally using the tip of a 30-gauge needle. A pulled glass micropipette with a tip diameter of approximately 40 µm, pre-filled with mineral oil and the AAV solution (Wiretrol II 1–5 µL, pulled with a Sutter Instrument P77, Novato, CA, USA), was introduced into the parenchyma of the left or right DVC using a micromanipulator. The NTS targeting coordinates were determined relative to the obex: rostro-caudal +0.1 mm, medio-lateral ±0.4 mm, and dorso-ventral −0.35 mm. After a two minutes wait, 300 nL of the viral vector solution was injected progressively in 6 injections of 50 nL, spaced 30 s apart, at a rate of 10 nL/s. The micropipette was left in place for an additional two minutes post-injection, then gradually withdrawn in one increment per minute for 2 min. After the injection, the incision was closed with one or two sutures. The mice received an intraperitoneal injection of carprofen (5 mg/kg, Zoetis, Châtillon, France) and of a rehydration solution (0.9% NaCl). They were then transferred to a heated recovery cage and monitored until fully awake. Analgesic treatment was provided for three days after surgery to ensure proper recovery. Body weight, general behavior, and wound healing were regularly monitored for two weeks following injection until full recovery.

### 2.3. Indirect Calorimetry and Metabolic Cages

At 6- or 7-weeks post-injection (WPI), mice were housed individually in Oxylet Physiocage metabolic cages (Panlab/Harvard apparatus, Cornellà, Spain), with the room maintained at a temperature of 25–27 °C. Mice were acclimated to the environment during the first 24-h light-dark cycle. Energy balance parameters were measured over the subsequent 48 h. The physiocages were equipped with a laser absorption sensor to measure oxygen (O_2_) levels, an infrared sensor for carbon dioxide (CO_2_) levels, and an open-circuit calorimeter for indirect calorimetry. Ambient air was circulated through each chamber at a flow rate of 400 ± 10 mL/min. O_2_ and CO_2_ levels were recorded every 30 min, with one measurement per second for 3 min during each sampling period. The O_2_/CO_2_ analyzer was calibrated daily at the end of the light phase using purified gas standards (high O_2_ and CO_2_ gas: 50.2% O_2_, 3.99% CO_2_; low O_2_ and CO_2_ gas: 20.02% O_2_, 0% CO_2_). Data were analyzed using METABOLISM V2.2.01 software (Panlab). Oxygen consumption (VO_2_) and carbon dioxide production (VCO_2_) were expressed in mL/min/kg^0.75^. The respiratory exchange ratio (RER) was calculated as the ratio of VCO_2_ to VO_2_. Energy expenditure (EE) for each animal (kcal/day/kg^0.75^) was determined using the formula: EE = VO_2_ × 1.44 × [3.815 + (1.232 × RER)]. The physiocages also had weight sensors to continuously monitor food and water intake. Mice were weighed, and feeders were replenished daily at the end of the light phase. Food intake for mice fed the NC diet was measured every 30 s, synchronized with the circadian cycle, starting at ZT12 (the beginning of the dark phase). A “bout” was defined as a period of food consumption during which the weight decreased by at least 0.2 g compared to the previous measurement. Bouts were separated by at least one measurement with no consumption. Meals were defined as consisting of one or more bouts, with an interval of at least 5 min between meals.

### 2.4. Oral Glucose Tolerance Tests (OGTT)

OGTTs were conducted during the 6th and 7th WPI on animals habituated to handling and oral dosing for 8 days. Before each OGTT, animals were fasted for 14 h (with water available *ad libitum*). D-glucose solution (1.5 g/kg body weight, dissolved in drinking water) was administered orally. Blood glucose levels were measured by collecting a drop of blood from the tip of the tail and applying it directly to a glucose test strip, which was analyzed using a glucometer (Accu-Chek Performa Nano, Roche Diagnostics Corporation, Indianapolis, IN, USA). Blood samples were collected immediately before glucose administration and at 5, 20, 60, 90, 120 and 240 min after the glucose bolus.

### 2.5. Immunohistochemistry

Mice were given buprenorphine subcutaneously (0.05 mg/kg) for analgesia before being anesthetized by intraperitoneal injection of ketamine (120 mg/kg) and xylazine (16 mg/kg). In some groups, intracardiac blood sampling was performed before the intracardiac perfusion (see Section 2.8). The perfusion procedure involved administering 2.5 mL of 0.1 M phosphate-buffer saline (PBS) at pH 7.4 containing 10% heparin (Choay, Cheplapharm, Levallois-Perret, France), followed by cold 0.1 M PBS, and then cold 4% paraformaldehyde (PFA) in 0.1 M PBS. The brains were immediately removed, post-fixed in 4% PFA for 1 h at room temperature, and then rinsed overnight in PBS. They were cryoprotected for 24–48 h at 4 °C in a 30% sucrose solution prepared in PBS. The brains were then frozen by immersion in isopentane at −40 °C, and coronal or sagittal sections (40 µm thick) containing the DVC were obtained using a cryostat (Leica CM3050, Nanterre, France) and collected serially in 0.1 M PBS. Some tissue sections were preserved in an antifreeze solution (Ethylene glycol and glycérol based cryoprotectant) before proceeding with the immunohistochemistry (IHC) protocol. Floating-section immunostaining was performed for Cx43, GFAP, EGFP, and Ionized Calcium Binding Adaptor Molecule 1 (IBA1). Sections were incubated for 1 h in a blocking solution: PBS 1× containing 0.3% Triton X-100 and 3% normal goat serum (for GFAP and EGFP), 3% horse serum (for IBA1), or 5% normal goat serum combined with 1% bovine serum albumin (for Cx43). Primary antibodies were applied as follows: anti-Cx43 (1:3000, rabbit, #ACC-201, Alomone Labs, Jerusalem, Israel), anti-GFAP (1:1000, mouse, #G3893, Sigma Aldrich, St. Louis, MO, USA), anti-EGFP (1:1000, mouse, #ab-38689P, Abcam, Cambridge, UK), and anti-IBA1 (1:500, goat, #ab5076, Abcam, Cambridge, UK). After overnight incubation at room temperature, the sections were labeled with corresponding secondary antibodies conjugated to Alexa Fluor dyes: Alexa Fluor 488 (for EGFP, #A-11029, Invitrogen, Carlsbad, CA, USA), Alexa Fluor 594 (for Cx43, GFAP, or IBA1, #A-11012, #A-11058, Invitrogen), and Alexa Fluor 405 (for GFAP, #A-11012, Invitrogen). Secondary antibodies were used at a dilution of 1:400 for 2 h at room temperature. After washing in PBS, sections were mounted on gelatin-coated slides and coverslipped with Mowiol mounting medium.

### 2.6. Image Acquisition and Data Processing

Fluorescence images were acquired using a Zeiss LSM 710 confocal microscope (Zeiss, Jena, Germany) and Zen 3.12 software. For double-labeling experiments, images were acquired sequentially. In some cases, mosaics of images at ×10 magnification were created to visualize the entire section with high definition. All images were processed using Adobe Photoshop CS6, limited to contrast and brightness adjustments.

Quantification of Cx43 immunoreactivity: fluorescent images of Cx43 labeling at different anteroposterior levels of the DVC were acquired using Z-stack mode. The laser 594 nm and detector settings were maintained consistently across all images. To clearly visualize the entire section, image projections from different focal planes were generated using ZEISS ZEN 3.12 software. Cx43 immunoreactivity was quantified using ImageJ Fiji 1.54f software (URL accessed 7 March 2023; https://imagej.net/software/fiji). In ImageJ Fiji, the color channels were separated, and images were converted to grayscale before saving in 8-bit TIF format. Brightness and contrast were adjusted, followed by converting images to binary using the Image|Adjust|Threshold function. The intensity of Cx43 labeling was quantified within the defined DVC regions using the Analyze Particles function. Results were expressed as a percentage relative to the Cx43 labeling observed in AAV-shRNA-Scramble animals.

Analysis of astrocytes morphology: fluorescent images of GFAP labeling were acquired in Z-stack mode to capture the complete astrocyte arborization. The laser and detector settings were kept constant for all images to ensure comparability. A maximum intensity projection of different focal planes was then generated to create a single image. To evaluate GFAP labeling under different conditions, analyses were performed using ImageJ Fiji 1.54f software (NIH, Bethesda, MD, USA). Because GFAP staining intensity was higher in cell bodies compared to the processes, brightness and contrast were adjusted to enhance the visualization of all astrocyte processes. Color channels were removed, contrast was inverted, and images were binarized to depict cells and their processes in black. Using thresholding and size filtering functions, small dark particles (<14 µm^2^) were excluded from analysis. The number of GFAP+ cells in each region of interest and the total area covered by GFAP+ signal (µm^2^) within these regions were quantified using the “Analyze” function in ImageJ Fiji 1.54f.

Morphological analysis of microglial cells: A Z-stack of 8 images, covering a 15 µm thickness of IBA1 staining, was acquired using a 40× objective lens. Microscope settings were kept consistent across all animals. The images were then combined into a single projection using the Maximum Intensity Projection function in ZEISS ZEN software. Morphological analysis was performed using ImageJ Fiji 1.54f software (URL accessed 7 March 2023, https://imagej.net/software/fiji), utilizing the AnalyzeSkeleton (2D/3D) and Neuroanatomy plugins as described by Young and Morrison [27]. These plugins facilitated efficient analysis of microglial branching. In ImageJ Fiji, color channels were separated, and images were converted to grayscale before being saved in 8-bit TIF format. Brightness and contrast adjustments were made to enhance visualization of all microglial processes using the Image|Adjust|Threshold function. The image was then converted to binary format using Process|Binary|Convert to Mask. Noise reduction was applied with the Process|Noise|Despeckle function. Further image cleaning was performed using the Process|Noise|Remove Outliers and Process|Binary|Close functions, where the Remove Outliers function connected dark pixels separated by up to two pixels, and the Close function removed outliers. The cell skeletons were generated using the Skeletonize (2D/3D) plugin and analyzed with the AnalyzeSkeleton (2D/3D) function. This analysis provided metrics on cell arborization, including the number of branches, junctions, average branch length, total branch length, and arborization area. To further assess arborization, Sholl analysis was performed to quantify microglial branching density by determining the number of intersections within the processes [28]. The Neuroanatomy/Sholl/Sholl Analysis (From image) plugin was used for this purpose.

### 2.7. Real-Time PCR Analysis

Mice were anesthetized with a mixture of ketamine (120 mg/kg) and xylazine (16 mg/kg), administered intraperitoneally (10 mL/kg) at 9 WPI, and then sacrificed by decapitation. The brains were extracted, and the DVC was collected, rinsed in cold artificial cerebrospinal fluid (aCSF: 87 mM NaCl, 25 mM NaHCO_3_, 75 mM sucrose, 10 mM glucose, 2.5 mM KCl, 1 mM NaH_2_PO_4_, 7 mM MgCl_2_, and 0.5 mM CaCl_2_), and then snap-frozen in liquid nitrogen before being stored at −80 °C. These samples were used for transcript expression analysis by RT-qPCR. mRNA extraction and reverse transcription (RT) were performed as described previously [29]. Briefly, total RNA was extracted from frozen tissues using TRI Reagent^®^ (#TR118, Molecular Research Center, Cincinnati, OH, USA) following the manufacturer’s instructions. RT was conducted using M-MuLV Reverse Transcriptase with random hexamer primers and oligod(T)20 from the OneScript^®^ OZY012 kit (Ozyme, Saint-Cyr-l’École, France). Gene expression analysis by real-time PCR was performed using the QuantStudio 7 Cycler (Applied Biosystems, Waltham, MA, USA) with SYBR Green Master Mix (#A25742, PowerUp, Thermo Fisher Scientific, Vilnius, Lithuania). The equivalent of 10 ng of initial RNA was subjected to PCR amplification in a final volume of 10 μL using 1 μM specific primers (Appendix A). Specific PCR product generation was confirmed by melting-curve analysis. Tyrosine 3-monooxygenase/tryptophan 5-monooxygenase activation protein zeta (YWHAZ) was used as the internal reference gene.

### 2.8. Plasma Assays

Blood samples were collected via intracardiac puncture before perfusion (see Section 2.5) using syringes with 16-gauge needles pre-treated with heparin (Choay, Cheplapharm, Levallois-Perret, France). The blood was transferred to microtubes containing 10 µL of heparin, and then centrifuged at 3000× *g* for 15 min at 4 °C to separate plasma. Plasma leptin and insulin levels were measured using mouse-specific ELISA kits (#90030 and #90080, respectively; Crystal Chem, Elk Grove Village, IL, USA) following the manufacturer’s instructions. The optical density was respectively measured at 450 nm and 630 nm, leptin and insulin levels were calculated from a standard curve.

### 2.9. Glycated Proteins

Glycated serum proteins (GSP) were assessed using a quantitative kit which determined glycated serum proteins in mouse plasma. The Mouse GSP kit uses proteinases to digest serum proteins into low molecular weight glycated protein fragments and uses specific fructosaminase to yield glucosone and H_2_O_2_. The H_2_O_2_ released is measured by a colorimetric reaction according to the manufacturer’s instructions. The absorbance at 570 nm is proportional to the concentration of glycated serum protein (μmol/L). Plasmatic GSP levels were measured using a commercial kit (Crystal Chem, Elk Grove Village, IL, USA).

### 2.10. Statistical Analysis

All results are presented as mean ± SEM. Statistical analysis was performed using GraphPad-Prism v 6.0. Cx43 immunofluorescence was compared to an unpaired, two-tailed Student’s *t*-test. All remaining endpoints were addressed with ANOVA models: a two-way ANOVA (viral construct × diet) followed by Tukey’s post hoc test was applied to astrocytic variables, microglial skeleton metrics, Sholl-derived arborization indices, non-longitudinal metabolic read-outs (body-weight distributions, mean respiratory quotient and energy expenditure), c-Fos-positive cell counts and RT-qPCR data. Longitudinal outcomes, including weekly body-weight trajectories, blood glucose levels during the OGTT or 2-DG challenge, cumulative food intake and time-resolved respiratory-quotient traces, were analyzed by two-way repeated-measures ANOVA. Multiplicity was controlled with Tukey’s or Bonferroni post hoc tests. Statistical significance was accepted at *p* < 0.05, and figures encode significance levels as */# (*p* < 0.05), **/## (*p* < 0.01), ***/### (*p* < 0.001) and ****/#### (*p* < 0.0001).

## 3. Results

### 3.1. Injection of AAV-ShRNA Through the Atlanto-Occipital Approach Strongly Reduces Cx43 Expression in DVC Astrocytes

Immunohistochemistry revealed remarkably strong Cx43 expression within the DVC especially in the NTS and DMNX while it was less abundant in the AP (Figure 1A). This Cx43 expression distinguished the DVC from other bulbar regions (Figure 1A). Moreover, as expected, Cx43 staining was found strongly associated with GFAP+ protoplasmic astrocytes (Figure 1B–D). To reduce Cx43 expression in adult mice DVC astrocytes and assess the impact on energy homeostasis, we used short hairpin RNA (shRNA)-based interference. These hairpin RNAs were specifically designed to target Cx43 (shRNA-Cx43) or to serve as an inactive control (shRNA-Scramble). We used viral vectors carrying RNA interference sequences under the control of the GFAP promoter to specifically target astrocytes and tanycyte-like cells of this structure [30]. This promoter also allowed for the expression of EGFP, which served as a marker for vector incorporation. Furthermore, the shRNA-Scramble and shRNA-Cx43 vectors were encapsulated in AAV-DJ8, a hybrid adeno-associated viral vector formed by combining capsid sequences from several AAV serotypes (including AAV2 and AAV8). AAV-DJ8 vectors under the GFAP promoter are known for their increased efficiency in transducing glial cells, particularly astrocytes [31,32]. We developed a surgical technique using the atlanto-occipital approach to precisely target the NTS, using the obex as an anatomical landmark. Injections were performed directly into the parenchyma of the NTS under stereotaxic control. We performed bilateral injections of an AAV solution into the NTS to visualize the incorporation of our viral vectors and evaluate their ability to selectively reduce Cx43 expression in GFAP+ glial cells of the DVC within a timeframe suitable to assess the impact on energy homeostasis (Figure 2A). Under our injection conditions, EGFP fluorescence was visible along the rostrocaudal axis of the NTS and DMNX as early as 3 weeks post-injection and was maintained up to 9 weeks post-injection (Figure 2B–H). This 3-week delay is consistent with literature describing the time required for viral vector incorporation and expression [33]. EGFP was associated with GFAP-expressing glial cells in the NTS and DMNX parenchyma, confirming the glial tropism of the AAVs and their expression by astrocytes (Figure 2C,E). Furthermore, EGFP was not found associated with IBA+ microglia (Figure 2F) or NeuN+ neurons (Figure 2G). Following Cx43 immunostaining, we observed a significant reduction in this protein in areas showing EGFP fluorescence after AAV-shRNA-Cx43 injection, but not after AAV-shRNA-Scramble injection, indicating that viral vector expression was associated with Cx43 knockdown (KD, Figure 2B–E,I). Quantification of Cx43 staining intensity along the longitudinal axis of the DVC, 9 weeks post-surgery, confirmed the strong reduction (~50 to 80% according to the DVC level) of Cx43 expression in all parts of the structure (Figure 2I). Brains that did not properly integrate the AAVs were excluded from the study after retrospective examination. RT-qPCR analyses revealed a ~40 to 50% reduction in Cx43 expression in brainstem samples collected week 4 (Appendix A) and 9 (Figure 2J) after AAV injection. This reduction appeared slightly less pronounced than that observed via immunostaining, likely because brains with incorrect or partial AAV integration could not be excluded and the dissected region used for qPCR was significantly larger than the area incorporating the AAVs, given the small size of the DVC. Thus, Cx43 KD became efficient at the 4th week after AAV injection and plateaus until the week-9, allowing us to collected physiological readouts during this window. Interestingly, the KD of Cx43 was not associated with compensatory up-regulation in the expression of other connexins and pannexin 1. Nevertheless, reductions in Cx30 and Cx45 were observed (Figure 2J). The diet provided to the animals, i.e., normal chow (NC) or HFS diet did not significantly alter Cxs and pannexin 1 expression (Figure 2J).

### 3.2. Impact of Cx43 Knockdown on GFAP+ Astrocytes in the NTS/DMNX Under Standard and Obesogenic Feeding Conditions

We next investigated whether the glial compartment was affected by reduced Cx43 expression under both standard and obesogenic diets. To this end, we performed morphological analyses of GFAP+ astrocytes 9 weeks after virus injection (Figure 3A). At early stages of model characterization, we observed that areas with reduced Cx43 expression due to AAV-shRNA-Cx43 injection, were associated with increased GFAP fluorescence. This increase was visible in all AAV-transduced-Cx43 regions (Figure 3B,C). These changes in GFAP expression and astrocyte morphology were particularly visible in the NTS and DMNX of AAV-shRNA-Cx43 (Figure 3D–G). When the hypoglossal nucleus was targeted by AAV-shRNA-Cx43 a strong GFAP reactivity was also observed in this structure (Figure 3D). To further characterize the modifications of astrocytic phenotype induced by Cx43 KD, we performed an Image J analysis of the astrocytic morphology and cell number within the DVC (Figure 3E–G). This analysis revealed both a significant increase in the cell size (Figure 3F) and in the surface area of the NTS/DMNX covered by GFAP+ astrocytes (Figure 3G) in animals that received AAV-shRNA-Cx43 and maintained on the NC diet, compared to their AAV-shRNA-Scramble counterparts. As the number of GFAP+ cells was not increased in AAV-shRNA-Cx43 mice (Figure 3E), it is likely that astrocytes extended their processes individually. These observations suggest a morphological adaptation of astrocytes in response to reduced Cx43 expression. We next sought to determine whether long-term exposure (12 weeks) to HFS diet could modify astrocytic morphology and/or GFAP expression in the DVC *per se* or modify astrocytic plasticity induced by Cx43 KD. Some recent studies have reported that GFAP+ astrocytes in rodent NTS can undergo morphological changes and modification of GFAP expression following acute [15,34] or prolonged [9] exposure to a fat-enriched diet. However, long-term effects of a HFS diet on glial morphology remain unexplored. In AAV-shRNA-Scramble_HFS-treated mice neither cell number nor size nor GFAP+ cell-covered surface area of NTS/DMNX were modified in comparison to AAV-shRNA-Scramble mice fed with NC diet; Figure 3E–G). Although GFAP does not detect all astrocytic dynamics, these results suggest that astrocytes morphology and density remain stable despite the metabolic stress associated with the HFS diet. In contrast, AAV-shRNA-Cx43_HFS animals showed significantly increased numbers of GFAP+ cells in the NTS/DMNX compared to AAV-shRNA-Cx43_NC (Figure 3E). Whereas GFAP-covered surface area is similarly large in AAV-ShRNA-Cx43 fed a NC or a HFS diet (Figure 3G), HFS-fed AAV-ShRNA-Cx43 mice do not display the high astrocytic size observed in AAAV-ShRNA-Cx43 fed a NC diet (Figure 3F). Thus, while the HFS diet did not alter astrocytic morphology in AAV-shRNA-Scramble_HFS mice, it affected the number and size of GFAP+ astrocytes present in the DVC of AAV-shRNA-Cx43 mice. This suggests that either astrocytic proliferation or migration to the AAV-transduced region in the context of obesity and impaired astrocytic communication via Cx43. In addition, RT-qPCR analysis revealed a significant reduction in GLAST expression in AAV-shRNA-Cx43 treated animals regardless of the diet, while GLT-1 expression remained unchanged (Figure 3H). These findings suggest that Cx43 KD may affect glutamate homeostasis, as evidenced by changes in astrocytic glutamate transporter expression. Confirming this alteration in glutamatergic signaling, several metabotropic glutamate receptors also showed altered expression following Cx43 KD, further supporting a potential impact of reduced astrocytic Cx43 on glutamatergic signaling in the DVC (Figure 3H).

### 3.3. Astrocytic Cx43 Knockdown Induces Microglial Hyper-Ramification in the NTS/DMNX

Given the astrocytic morphological changes observed in AAV-shRNA-Cx43 treated mice, we hypothesized that the entire glial network could be remodeled. Astrocytes and microglial cells maintain close communication, notably through paracrine interactions involving the release of chemical messengers via HCs [35,36]. In pathological contexts, microglial and astrocytic reactivity often go hand in hand [37,38]. Based on this, we sought to determine whether, in our model of Cx43 KD, DVC microglial cells undergo morphological remodeling, and whether these effects varied as a function of a standard or obesogenic diet. To this end, we performed IBA1 immunostaining on caudal brainstem slices (Figure 4A,B). At low magnification, IBA1 fluorescence intensity appeared higher in the DVC parenchyma of AAV-shRNA-Cx43-treated animals, compared to those treated with AAV-shRNA-Scramble (Figure 4A). These preliminary observations suggest a reactivity of the microglial compartment in response to reduced Cx43 expression and/or alterations in astrocytic morphology. At the cellular level, microglia in AAV-shRNA-Cx43 mice exhibited more extensive and complex arborization (Figure 4B). To confirm these observations, we binarized IBA1 images and generated skeletons to enable morphometric analysis (Figure 4C–G). These quantitative analyses revealed that microglia from AAV-shRNA-Cx43-treated mice exhibited an increase in the number of branches (Figure 4D) and a ~67% increase in total branch length (Figure 4E) compared to AAV-shRNA-Scramble. The area covered by individual cells was also increased by ~63% (Figure 4F). However, the average length of individual branches remained unchanged (Figure 4G). These results indicated that, in response to decreased astrocytic Cx43 expression, microglial cells undergo significant morphological remodeling, adopting a more complex and ramified arborization. Sholl analysis is now a well-described and widely used approach to quantify microglial cell arborization complexity and study their plasticity [28]. This method allows the study of the arborization density of a cell by quantifying the number of intersections between extensions as a function of distance from the soma (Figure 4H–L). We observed an increase in the number of intersections all along the arborization, from 15 µm to the furthest distances from the soma in mice treated with AAV-shRNA-Cx43 when compared to those treated with AAV-shRNA-Scramble (Figure 4I). This is attested by the increase in the numbers of junctions (Figure 4J), and in triple and quadruple points, i.e., two or three branches emerging from a single initial extension, respectively (Figure 4K,L). This indicates increased branching of cell extensions throughout the structure, both in proximal and distal regions of the arborization. The extensions also extended at longer distances from the soma. Indeed, in AAV-shRNA-Scramble mice, intersections were present up to 45 µm from the soma, while in AAV-shRNA-Cx43 mice, they extended up to 80 µm (Figure 4I). We did not observe an effect of the HFS diet on microglial cell morphology. Indeed, Skeleton analysis and arborization complexity were similar in AAV-shRNA-Scramble mice regardless of diet (Figure 4D–L). In mice with reduced Cx43 expression, the nature of the diet did not alter cytoskeleton complexity or the ability of microglial cells to respond to the altered astrocytic compartment.

### 3.4. Reduction In Astrocytic Cx43 Expression Did Not Alter the Metabolic Phenotype of Mice

To assess the impact of Cx43 deletion on energy balance regulation, we performed longitudinal monitoring of body weight gain in animals that received either AAV-shRNA-Scramble or AAV-shRNA-Cx43 at 8 weeks of age. Cx43-deleted animals and controls were fed either NC or HFS diet from weeks 5 to 7 (Figure 5A). Throughout the experiment, we monitored the animal’s weight gain with two weighings per week. No statistically significant differences in body weight growth were observed between AAV-shRNA-Scramble and AAV-shRNA-Cx43 groups fed with NC (Figure 5B–D). Under the HFS diet, the mice developed moderate obesity, reaching an average weight of 35–37 g after 12 weeks, compared to 28 g in mice maintained on NC (Figure 5B–D). As observed in NC cohorts, reduction in Cx43 expression did not significantly affect body weight gain or final body weight of mice maintained on HFS diet (Figure 5B–D). However, a tendency towards higher body weight under the HFS diet was observed in mice that received AAV-shRNA-Cx43 compared to those that received AAV-shRNA-Scramble. This trend, apparent at the 15th week of age, intensified until the 17th week of age, without reaching statistical significance (*p* = 0.0545, Figure 5B,C). This tendency was also visible in the distribution of body weights at the 12th week (Figure 5D). The mice on the HFS diet that received AAV-shRNA-Cx43 had the highest body weights, with a median of 36.9 g for AAV-shRNA-Cx43_HFS compared to 35.5 g for the AAV-shRNA-Scramble_HFS group (Figure 5D). Leptin levels were increased in both AAV-shRNA-Scramble_HFS and AAV-shRNA-Cx43_HFS mice compared to their NC-fed counterparts (Figure 5D inset). Although leptin levels were slightly higher in animals AAV-shRNA-Cx43_HFS mice, this difference was not statistically significant (Figure 5D inset). The longitudinal monitoring of the animals also included measurements of food rests over 24 h, allowing for the estimation of daily food intake. These measurements were performed between the 6th and 8th week after the AAV injections (equivalent to 14–16 weeks of age; Figure 5A). During this period, food intake of the mice was similar between the two groups submitted to the HFS diet, as well as between the two groups submitted to the NC diet, regardless of the viral vector administered (Figure 5B inset). Indirect calorimetry was then carried out to assess respiratory quotient (RQ) and energy expenditure in the four groups. The recordings revealed an increase in oxygen consumption associated with a decrease in RQ during the nocturnal phase in animals subjected to the HFS diet (Figure 5E,F). These results indicate that the prolonged HFS diet was associated with an increased use of lipids as an energy source. This decrease in RQ, classically described in diet-induced obesity models, is often associated with insulin resistance leading to an increased dependence on lipids [39]. The decrease in RQ was, however, similar between the HFS mice that received AAV-shRNA-Scramble and those treated with AAV-shRNA-Cx43, suggesting that the reduction in Cx43 expression does not alter the propensity to develop signs of metabolic syndrome. Similarly, energy expenditures were not modified by the reduction in astrocytic Cx43 expression (Figure 5G).

The fasting-refeeding protocol assesses fasting-induced orexigenic tone and integration of satiety signals after a 3-h refeeding period. Fasting-refeeding experiments were performed 9 weeks after AAV injections (Figure 6A). Body weight variation induced by 21 h of fasting and food intake during the 3 h refeeding period were comparable between the AAV-shRNA-Scramble and AAV-shRNA-Cx43 groups (Figure 6B,C). Additionally, c-Fos immunostaining revealed a significant increase in the number of c-Fos^+^ cells in the postremal (*p* = 0.0307) and rostral (*p* = 0.0479) parts of the NTS after 21 h fast followed by 3 h refeeding in mice from the AAV-shRNA-Cx43 group, compared to those from the AAV-shRNA-Scramble group (Figure 6D,E). In the AP and caudal NTS, cellular activation induced by refeeding after fasting was not significantly different between the two groups. These results suggest that Cx43 KD may alter neuronal response in specific subregions of the NTS following refeeding. Sensitivity to satiety signals may be heightened when glial communication via Cx43 is impaired.

### 3.5. Reduction In Astrocytic Cx43 Expression Did Not Alter Glycemic Regulation

In parallel with the study of energy metabolism, we assessed glycaemic regulations in animals with reduced astrocytic Cx43 expression that were maintained either on NC diet or HFS diet (Figure 7A). This choice was motivated by two considerations: first, the permeability of Cx43 HCs to glucose suggests a potential role in central glucosensing; and second, Allard et al. [23] provided the first evidence that astrocytic communication via Cx43 is involved in hypothalamic glucodetection. Weekly blood glucose measurements were performed on animals aged 10 to 16 weeks (2nd to 8th week after surgery, Figure 7B). Substantial variability was observed between the groups, probably due to the lack of fasting before measurements, although sampling was carried out during the middle of the light phase, a period where the mice consume little food. Despite this variability, a slight increase in baseline blood glucose was observed in mice fed an HFS diet compared to those maintained on a standard diet, indicating a diet-associated prediabetic state. No effect of reduced Cx43 expression on blood glucose levels was detected (Figure 7B). At 6- or 7-weeks post-injection, OGTTs were performed to assess the mice’s ability to metabolize glucose and regulate blood glucose levels after oral glucose bolus. The nature of the injected AAV had no effect on glucose tolerance: both groups fed the NC diet (i.e., AAV-shRNA-Scramble and AAV-shRNA-Cx43) exhibited an increase in blood glucose induced by D-glucose within the first 20 min after oral administration, followed by a gradual return to normoglycemia (Figure 7C). These results indicate that reduced Cx43 expression in the DVC did not disrupt blood glucose regulation after an oral glucose loading in mice on a NC diet. For both groups fed an HFS diet, compared to the groups maintained on NC, a more pronounced increase in blood glucose was observed 20 min after oral glucose administration, with a slower return to normoglycemia (Figure 7C). This glycemic profile is characteristic of glucose intolerance and may reflect reduced insulin sensitivity. Intriguingly, glucose intolerance appeared to be slightly more pronounced in mice fed an HFS diet receiving AAV-shRNA-Cx43 compared to those injected with AAV-shRNA-Scramble, although this difference was not statistically significant (Figure 7C). Indeed, the blood glucose measured 20 min post-gavage was slightly higher in the AAV-shRNA-Cx43_HFS mice (371.5 mg/dL) compared to the AAV-shRNA-Scramble_HFS mice (319.3 mg/dL). At 90 min post-gavage, blood glucose remained elevated in mice that received AAV-shRNA-Cx43_HFS (217.5 mg/dL) compared to those receiving AAV-shRNA-Scramble_HFS (175 mg/dL). These observations suggest that reduced Cx43 expression could slightly potentiate the development of glucose intolerance induced by an obesogenic diet. To further explore metabolism, we measured plasma insulin concentrations at 9 weeks post-surgery (12 weeks of diet). These analyses confirmed signs of metabolic syndrome in mice fed the HFS diet, indicating an adequate response to the diet for both groups of animals involved. Indeed, as expected, plasmatic insulin levels increased in animals fed with HFS diet. Interestingly, plasma insulin level was significantly higher in animals exposed to 12 weeks of HFS diet and receiving AAV-shRNA-Cx43 than in their counterparts fed with NC diet (Figure 7D). This increase in insulin levels suggests that, in the context of an obesogenic diet, the reduction in Cx43 expression in DVC astrocytes can potentiate the onset of metabolic syndrome symptoms, such as insulin resistance. Nevertheless, quantification of plasma glycated proteins in all experimental groups did not reveal an increased level of glycated protein in AAV-shRNA-Cx43 fed with HFS diet. Finally, we also sought to determine whether the response to glucose deprivation was modified by decreased Cx43 expression in the DVC. These counter-regulations are known to recruit neural networks from the DVC, and astrocytes in this region have been described as glucose-sensitive cells capable of activating neural networks involved in these emergency responses [40,41,42,43]. Glucoprivation was induced by intraperitoneal injection of 2-deoxyglucose (2DG, 0.3 g/kg, 10 mL/kg). We measured blood glucose levels, food intake, and cellular activation in response to 2DG-induced glucoprivation (Figure 8A). 2DG injection caused a sharp rise in blood glucose, which was similar in both AAV-shRNA-Scramble and AAV-shRNA-Cx43 groups (Figure 8B), indicating that Cx43 inhibition does not affect the glycemic response to glucoprivation. Food intake is known to increase significantly within three hours of administration of non-metabolizable glucose analogs. Under our experimental conditions, food intake increased similarly in both AAV-shRNA-Scramble and AAV-shRNA-Cx43 groups (Figure 8C). To assess cellular responses, we performed c-Fos immunostaining 2 h after 2DG or NaCl injection (Figure 8D). No significant differences in c-Fos expression were observed between groups (Figure 8E), suggesting that Cx43 inhibition does not impair neuronal responses to glucoprivation and that glucose sensing remains intact even when astrocytic gap junction coupling is disrupted. We also performed additional immunostaining against tyrosine hydroxylase (TH), labelling catecholaminergic neurons known to be involved in the response to glucoprivation. Although we did not perform specific counting of c-Fos^+^ nuclei belonging to TH neurons, activation of this cell population appears similar between groups (Figure 8F).

## 4. Discussion

The DVC displays very high levels of astrocytic Cx43 expression, which clearly distinguishes it from other brainstem structures. This enrichment suggests intense astrocytic communication via Cx43 HCs and Cx43 GJs in this structure. To test its functional relevance, we selectively knocked down Cx43 in DVC astrocytes using shRNA AAVs and assessed the impact on energy balance. In response to this depletion, we observed strong phenotypic changes in both astrocytes and microglial cells, while no significant impact on the energy balance of the animals was detected.

### 4.1. Efficient Knockdown of Cx43 in DVC Astrocytes via an Atlanto-Occipital Route

Precise stereotaxic delivery to DVC is challenging because the caudal medulla lies far from conventional cranial landmarks (bregma and lambda). We therefore adapted the atlanto-occipital route described by Brierley et al. [26], using the obex as a reference, and bilaterally injected GFAP-driven AAV-DJ/8 vectors (AAV-DJ/8-GFAP::EGFP) to target astrocytes. This promoter is suitable for NTS astrocytes, as ~98% are GFAP-positive [44]. EGFP fluorescence remained confined to astrocytes throughout the NTS and DMNX, with no neuronal or microglial expression. In animals that received the Cx43 construct, immunoreactivity along the entire rostro-caudal axis dropped by ~80% between weeks 3 and 9, confirming efficient knockdown. The reduction in Cx43 transcript levels measured by RT-qPCR 9 weeks after surgery was more modest (~50%), likely underestimated. Although transduction was homogeneous along the NTS–DMNX axis, it remained scarce in the AP and the *funiculus separans*. This outcome reflects the AP’s low proportion of GFAP-expressing cells [30] and mirrors previous reports of its poor stereotaxic accessibility, particularly to target its glial cells [45,46]. Importantly, the limited AP coverage is inconsequential for our objectives, because, like GFAP, Cx43 is only marginally expressed in the AP [22].

We next asked whether other connexins were transcriptionally upregulated to compensate for Cx43 KD. RT-qPCR revealed no up-regulation of alternative connexins. Instead, Cx30, the second most abundant astrocytic connexin in the DVC [22], declined in parallel with Cx43. This co-reduction appears specific to our model, as Cx43 genetic deletion in the hippocampus, cortex, and cerebellum typically has been associated with Cx30 overexpression [47,48,49]. Cx45 levels likewise fell, whereas Cx26 was unchanged. Given that Cx26 is expressed ~100-fold less than Cx43 and that reduced expression of Cx30 has been associated with destabilization of connexons consisting of Cx26 [50], persistent astrocyte coupling seems highly improbable in our model. Connexins restricted to microglia or oligodendrocytes (Cx32, Cx36, Cx47) were similarly unaffected. Altogether, in our model, the reduction in Cx43 expression was not accompanied by a molecular signature of compensatory pathways, enabling restoration of hemichannel and/or intercellular channel function.

### 4.2. DVC Astrocytic Cx43 Loss Does Not Disrupt Metabolic Balance

Over a 9-week period, body-weight gain, energy expenditure assessed by indirect calorimetry, and both *ad-libitum* and refeeding intakes were indistinguishable between the AAV-shRNA-Cx43 and AAV-shRNA-Scramble groups. Glycemic homeostasis, assessed by OGTT and the response to 2-DG-induced cellular glucose deprivation, were also preserved. The only detectable change was an increased number of c-Fos-positive neurons 3 h after refeeding, implying that Cx43-containing HCs and GJs might attenuate the satiety signal generated upon refeeding. Because a standard diet might be too permissive to reveal latent defects, we placed a second cohort on an HFS diet. Mice started the HFS diet at 5 weeks of age and received the viral vectors 3 weeks later, leaving a 9-week observation window that fits well with the period showing AAV-induced Cx43 KD. It should be noted that these constraints forced us to carry out the surgery after 3 weeks of HFS feeding and therefore on an already altered metabolic background. The HFS diet was introduced in young animals shortly after weaning to enhance its effects [51]. Even under this caloric overload, Cx43 depletion produced no significant effects on weight gain, energy expenditure, or glucose tolerance. Only subtle trends emerged, including a slightly heavier body mass (*p* ≈ 0.055), a larger glycemic excursion after glucose load, a delayed return to normoglycemia, and higher fasting insulin levels at week 9, suggesting a modest predisposition to obesity and insulin resistance that remains effectively compensated during our project. The possibility that a more pronounced phenotype may appear beyond 9 weeks after surgery can legitimately be considered.

The lack of a noticeable effect of Cx43 KD on the metabolic phenotype under both NC and HFS diets is puzzling. At first glance, this might suggest that astrocytic communication through Cx43 is irrelevant for the DVC’s metabolic functions. Although this cannot be excluded, it is surprising for at least three reasons. First, our data indicate that Cx43 is among the most abundant connexins in the DVC [22], suggesting that a large part of astrocytic coupling relies on this isoform. Second, current evidence provides several arguments for expecting a role of DVC astrocytes in energy and glucose control (see [7,8] for review), chemogenetic activation of DVC astrocytes reduces meal size [52], and it has been reported that they participate in the control of glycemia [40,41,43]. Indeed, Rogers and colleagues reported that a modest decrease in extracellular glucose activates this glia, whose purine production activates tyrosine hydroxylase-positive neurons in the NTS [43]. Third, ATP has been identified as an important gliotransmitter in the DVC circuits that integrate vagal afferent input. In acute slices, for example, Accorsi-Mendonca et al. [53] reported that inhibition of astrocytic metabolism or purinergic signaling decreases the amplitude and frequency of solitary-tract-evoked EPSCs onto NTS neurons. They elegantly identified ATP, released by NTS astrocytes upon solitary tract stimulation, as the gliotransmitter acting on the purinergic P2 receptor to stimulate presynaptic glutamate release. The involvement of Cx43 GJs and HCs in astrocytic calcium signals and ATP release is also widely documented [54]. We therefore hypothesized that convergence of Cx43 abundance, astrocyte-dependent modulation of feeding circuits, and ATP-mediated synaptic facilitation would normally predict at least subtle metabolic perturbation when Cx43 is deleted.

This observation is even more unexpected considering our previous study: intracerebroventricular delivery of the Cx43 HC blocker TAT-Gap19 induces a robust anorexigenic effect and rapidly activates DVC and hypothalamic neurons [22]. Nevertheless, we can consider several non-mutually exclusive explanations that may account for this discrepancy. The first concerns time course. AAV-mediated knockdown develops gradually, providing the network ample opportunity to recruit compensatory mechanisms. In contrast, TAT-Gap19 produces an abrupt blockade that may outpace adaptation. Consistent with this idea, inhibition of Cx43 hemichannels by TAT-Gap19 did not cause the astrocytic and microglial remodeling observed after chronic shRNA knockdown [22]. The second concerns channel selectivity. TAT-Gap19 exclusively targets HCs, whereas shRNA reduces total Cx43 protein, thereby abolishing both HC and GJ channels. Because these two conduits function in distinct physiological contexts, with HCs mediating transmitter release and GJ mediating metabolic and ionic coupling, this dual loss might trigger adaptive rewiring that is very different from selective hemichannel blockade (for review, see [55]).

### 4.3. Astrocytic Cx43 Depletion Induces Glial Compartment Remodeling

Under both the NC and HFS diets, we observed a marked expansion of GFAP-positive territory. On chow, this involved hypertrophy of existing astrocytic processes, whereas on HFS, this was mainly due to an increased number of GFAP-expressing cells. The diet-dependent nature of these changes argues against a non-specific surgical stress response and instead suggests that DVC astrocytes adapt their architecture to preserve network function when Cx43-mediated coupling is lost. Such plasticity is rarely reported; most lesion models associate astrogliosis with connexin upregulation (for review, see [56]). A plausible trigger is the disruption of inter-astrocytic K^+^ and glutamate buffering normally supported by Cx43 GJs. In the hippocampus, Wallraff et al. [57] reported that dispersed astrocytes depend more on gap-junctional networks for K^+^ redistribution than denser astrocyte networks. In our model, the denser alignment could help individual DVC astrocytes compensate locally when connexin scaffolding is weakened. Consistent with this idea, we detected altered expression of the glutamate transporter GLAST and several metabotropic glutamate receptors, especially after HFS exposure, suggesting a broader adjustment of glutamatergic signaling whose functional impact remains to be defined.

The microglial compartment was also remodeled. Although genetically preserved from our GFAP-driven vector, microglia exhibit longer and more numerous processes, leading to a highly branched, “hyper-ramified” phenotype. Cross-talk between the two glial populations may underpin this response. It is known that cytokine release by activated microglia enhances astrocytic Cx43 HC activity [58], while astrocyte–microglia co-culture alters Cx43 permeability and coupling [38,59]. In the present context, loss of astrocytic Cx43 could disrupt such bidirectional signaling and cause reciprocal structural plasticity. Traditionally, microglia with thin, ramified processes were considered “resting” [60], but accumulating evidence describes them as active sentinels whose branching complexity can increase under chronic stress [61], depressive-like behavior [62], or post-traumatic stress [63]. Recent work even links hyper-ramification to resilience against social defeat and to the expression of Arc/Arg3.1, a synaptic-plasticity marker [64]. Our data therefore fit a growing view that microglial branching is not simply a prelude to phagocytic activation but may facilitate long-range surveillance, synapse remodeling, and fine-tuning of inflammatory tone.

Several knowledge gaps remain. First, the functional consequences of expanded astrocytic coverage on K^+^ clearance and glutamate homeostasis in the DVC need to be tested electrophysiologically. Second, it is unclear whether microglial hyper-ramification plays a protective, pro-synaptic, or pro-inflammatory role here; transcriptomic profiling of isolated microglia would help clarify this point. Third, the trigger for the diet-specific astrocytic response remains to be identified. HFS diets raise circulating lipids and cytokines that can cross the blood–brain barrier, and both factors modulate astrocytic morphology. Disentangling the direct effect of Cx43 loss from diet-borne signals will require temporally controlled deletion combined with matched metabolic challenges. In summary, knocking down astrocytic Cx43 leaves systemic metabolism surprisingly unaltered but causes pronounced, bidirectional remodeling of astrocytes and microglia. These morphological adjustments likely may represent an intrinsic attempt to preserve local homeostasis when connexin-based communication is disrupted.

### 4.4. DVC Astrocytes: An Intrinsic Safeguard for Metabolic and Autonomic Control?

Our results reveal a striking paradox. Despite the exceptionally high basal expression of Cx43 in DVC astrocytes [22], its shRNA-mediated knockdown, accompanied by pronounced astro- and microgliosis, left body weight, glycemia, and indirect calorimetry unchanged. No other connexin was up-regulated, but neural output to satiety circuits appeared preserved. One plausible explanation is an intrinsic astroglial resilience that emerges during slow, chronic challenges but is overwhelmed by acute pharmacological blockade (e.g., TAT-Gap19; [22]). Such resilience would be advantageous in the DVC, a hub integrating energy and baro- and chemo-receptor information [65,66] while maintaining cardiorespiratory stability.

Evidence for adaptive plasticity in DVC glia is accumulating. Hypoxia, hypertension, or HFS diets increase GFAP content and remodel astrocyte morphology [15,67,68,69]. Pharmacological ablation of NTS astrocytes blunts baro- and chemoreflexes and precipitates cardiac arrhythmias [70], underscoring their autonomic importance. Notably, we detected shifts in GLAST and several mGluRs after Cx43 loss, changes consistent with reports that astrocytic glutamate handling shapes NTS output [12]. Microglial hyper-ramification observed here may further reinforce this compensation: cytokines such as IL-1β and TNF-α, released by activated microglia, amplify astrocytic Cx43 hemichannel activity [58] and could maintain residual purinergic signaling in the face of reduced gap-junction coupling [36,59].

## 5. Conclusions

Overall, our results show that disruption of astrocytic communication in the DVC via Cx43 deletion is accompanied by a strong astrocytic and microglial reaction. Astrocyte densification and microglial hyper-ramification could constitute an adaptive system capable of preserving vital functions operating at the brainstem level, such as metabolic and autonomic regulations, at least in the first few weeks. Although still hypothetical, this possibility warrants testing under longer and/or other conditions of autonomic dysfunction, such as more severe metabolic overload or prolonged hypoxia conditions [8].

## Figures and Tables

**Figure 1 cells-14-01694-f001:**
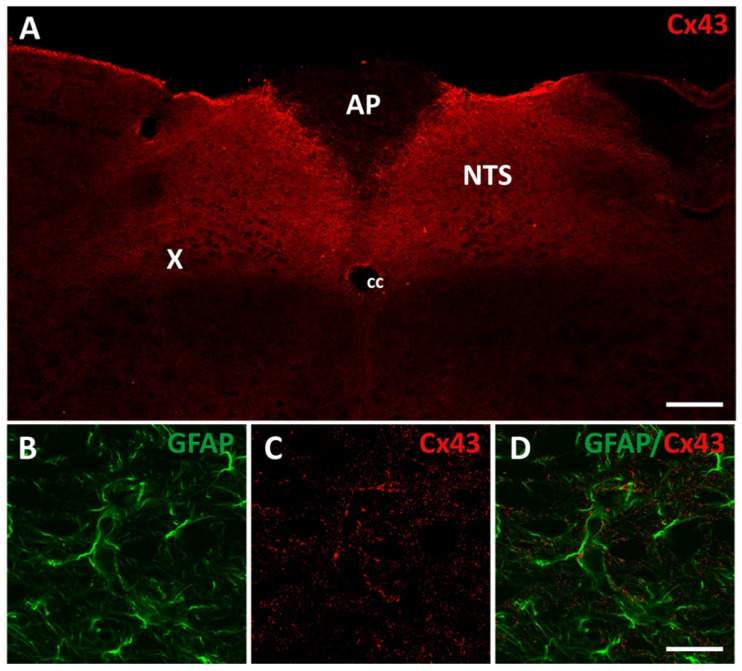
Cx43 expression within the DVC. (**A**) Confocal images of Cx43 immunofluorescence within the DVC. Note that Cx43 labeling is significantly more intense within the DVC than in surrounding brainstem structures. Scale bar: 200 μm. (**B**–**D**) Confocal images of double labeling for Cx43 (red) and GFAP (green) in the DVC. In the DVC, Cx43 is strongly associated with astrocytes GFAP+ (green) processes. AP: area postrema, NTS: nucleus tractus solitarius, cc: central canal; X: dorsal motor nucleus of the vagus nerve; GFAP: Glial Fibrillary Acidic Protein. Cx43: connexin 43. Scale bar: 20 μm.

**Figure 2 cells-14-01694-f002:**
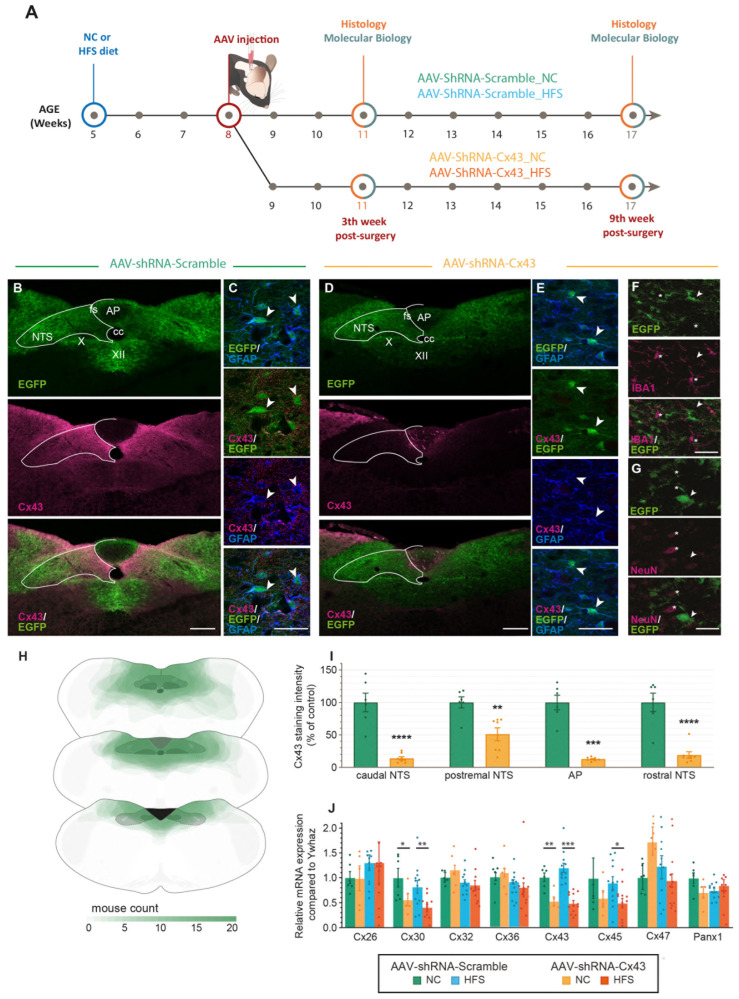
Knockdown of astrocytic Cx43 in the DVC. (**A**) Schematic illustration of the experimental timeline of AAV-shRNA injection. (**B**–**E**) Representative confocal images showing EGFP (green) and Cx43 (magenta) immunoreactivity in the DVC of mice treated with AAV-shRNA-Scramble (**B**,**C**) or AAV-shRNA-Cx43 (**D**,**E**). The white line delimits the DVC’s boundaries. Note the strong reduction in Cx43 labelling after AAV-shRNA-Cx43 injection. (**C**,**E**) Enlarged images of the NTS from panels (**B**,**D**), illustrating EGFP expression (green) in GFAP+ cells (blue). Arrowheads in (**C**,**E**) indicate GFAP+ cells that incorporated AAVs. AP: area postrema; Cx43: connexin 43; cc: central canal; EGFP: green fluorescent protein; fs: funiculus separens; GFAP: Glial Fibrillary Acidic Protein; X: Dorsal motor nucleus of the vagus nerve; XII: nucleus of the hypoglossal nerve; NTS: nucleus of the solitary tract. Scale bars: 300 µm (**B**,**D**), 50 µm (**C**,**E**) and 30 µm (**F**,**G**). (**F**,**G**) Representative confocal images of EGFP (green) and IBA1 ((**F**), magenta) or NeuN ((**G**), magenta) co-staining. Microglia or neurons (indicated by the asterisks) were not detected positive for EGFP. Arrowheads indicate GFAP+ cells that incorporated the AAVs and negative for IBA1 or NeuN, respectively. Asterisks indicate IBA1 or NeuN positive cells, negative for GFP. IBA1: ionized calcium-binding adaptor molecule 1, microglia marker; NeuN: neuron-specific nuclear marker. (**H**) Overlay of transduced regions analyzed 9 weeks after AAV injection. The most colored areas represent regions common to all mice (n = 20). (**I**) Quantification of Cx43 staining intensity along the longitudinal axis of the NTS and the AP, nine weeks after AAV injection. Data are presented as mean ± SEM, expressed as a percentage relative to mice treated with AAV-shRNA-Scramble. AAV-shRNA-Scramble: n = 7; AAV-shRNA-Cx43: n = 9. * Compared to mice receiving AAV-shRNA-Scramble: *p* < 0.01: **; *p* < 0.001: ***; *p* < 0.0001: ****. (**J**) RT-qPCR quantification of connexins (Cx) and pannexin 1 (Panx1) expression levels in the DVC, normalized by YwHAz gene, at nine weeks post-injection. Data are presented as mean ± SEM. AAV-shRNA-Scramble_NC (n = 6); AAV-shRNA-Cx43_NC (n = 5); AAV-shRNA-Scramble_HFS (n = 12); AAV-shRNA_Cx43_HFS (n = 13). * Compared to respective control, i.e., mice receiving AAV-shRNA-Scramble and fed NC or HFS: *p* < 0.05: *; *p* < 0.01: **; *p* < 0.001: ***.

**Figure 3 cells-14-01694-f003:**
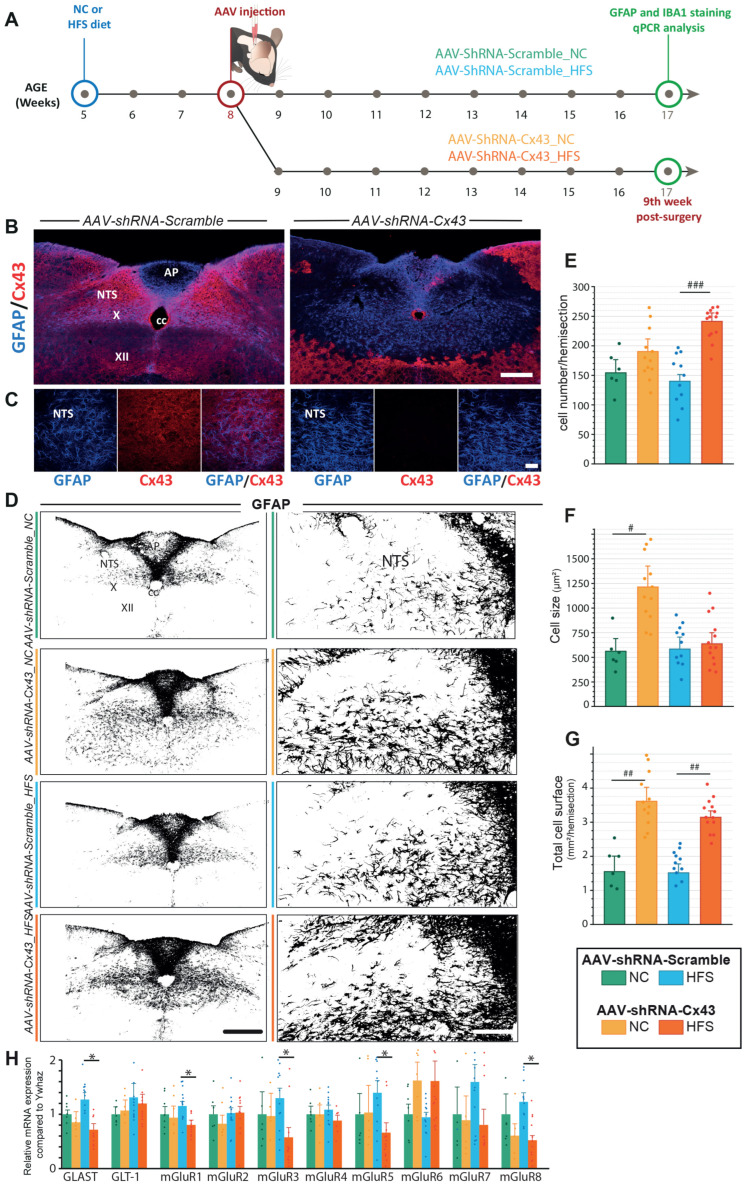
Impact of reduced Cx43 expression on DVC GFAP+ astrocytes morphology. (**A**) Experimental timeline. Mice were fed a standard NC or HFS diet from 5 weeks of age (i.e., 3 weeks before surgery). At 8 weeks of age, all mice received a bilateral injection of AAV-shRNA-Scramble or AAV-shRNA-Cx43 into the DVC. Histological analyses were performed on 17-week-old animals (9 weeks post-surgery and 12 weeks on diet). Animals excluded from post-mortem analysis, due to lack of complete AAV integration in the DVC, were not included in the results. (**B**,**C**) Representative confocal images of GFAP (blue) and Cx43 (red) immunostaining in the DVC. In AAV-shRNA-Scramble-treated mice ((**B**,**C**), **left**), Cx43 fluorescence intensity is uniform. In mice treated with AAV-shRNA-Cx43 ((**B**,**C**), **right**), the AAV-transduced area is clearly delineated by a reduction in Cx43 fluorescence intensity. Note that the regions lacking Cx43 labelling exhibited high GFAP fluorescence intensity. AP: area postrema; cc: central canal; X: motor nucleus of the vagus nerve; XII: nucleus of the hypoglossal nerve, NTS: nucleus of the solitary tract. GFAP: Glial Fibrillary Acidic Protein. Cx43: connexin 43. Scale bars: 300 µm (**B**) and 50 µm (**C**). (**D**) Representative binarized images of GFAP immunolabeling from the different experimental groups. Left panels: Low magnification images illustrating GFAP labelling in the DVC and surrounding brainstems structures. Right panels: High magnification of GFAP immunoreactivity within the NTS and DMNX. Scale bars: 300 µm (left panels) and 150 µm (right panels). (**E**–**G**) Morphological analysis of GFAP+ cells in the NTS/DMNX region. The mean number of GFAP+ cells (**E**), the average size of GFAP+ astrocytic extensions (**F**) and area of brain parenchyma covered by GFAP+ cells (**G**) were quantified for each experimental group: AAV-shRNA-Scramble_NC (n = 3, 6 sections); AAV-shRNA-Cx43_NC (n = 6, 12 sections); AAV-shRNA-Scramble_HFS (n = 6, 12 sections) et AAV-shRNA-Cx43_HFS (n = 7, 14 sections). Data are expressed as mean ± SEM. # compared to respective control, i.e., mice receiving AAV-shRNA-Scramble and fed NC or HFS: *p* < 0.05: #; *p* < 0.01: ##; *p* < 0.001: ###. (**H**) RT-qPCR quantification of glutamate transporters (GLAST and GLT-1) and metabotropic receptors (mGluR1-8) expression in the DVC, normalized by YwHAz gene, at nine weeks post-injection. Data are presented as mean ± SEM. AAV-shRNA-Scramble-NC (n = 6); AAV-shRNA-Cx43_NC (n = 5); AAV-shRNA-Scramble_HFS (n = 12); AAV-shRNA-Cx43_HFS (n = 13). * Compared to respective mice receiving AAV-shRNA-Scramble and fed HFS: *p* < 0.05: *.

**Figure 4 cells-14-01694-f004:**
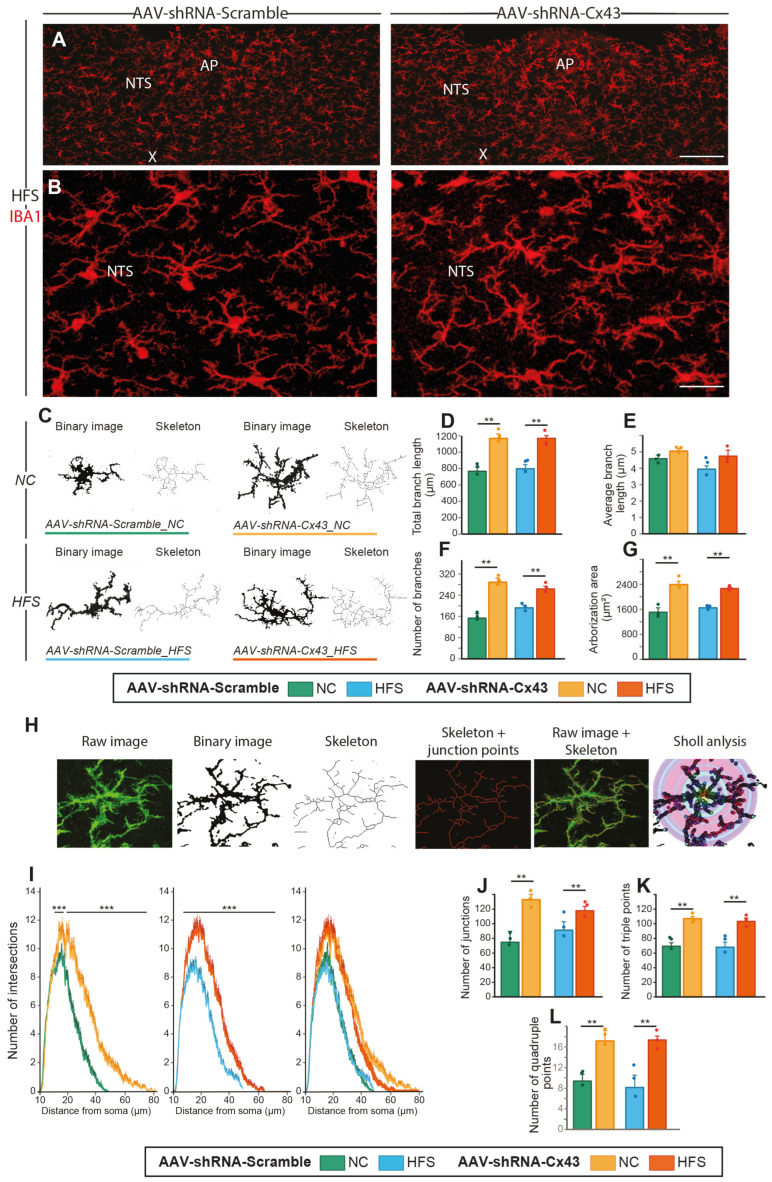
Impact of reduced Cx43 expression on DVC IBA+ microglia morphology. (**A**,**B**) Visualization of IBA1 immunoreactivity in the DVC of mice treated with AAV-shRNA-Scramble (left panels) or AAV-shRNA-Cx43 (right panels). Note the overall fluorescence intensity, which appears higher in the AAV-shRNA-Cx43 group (right panels). Images in (**B**) are magnifications of the images shown in A allowing for observation of the density of microglial cell processes. AP: area postrema; X: motor nucleus of the vagus nerve; NTS: nucleus of the solitary tract. Scale bars: 300 µm (**A**) and 50 µm (**B**). (**C**) Binary images obtained by IBA1 immunostaining and their representations as skeletons of microglia in the DVC. The images show individual microglial cells in mice treated with AAV-shRNA-Scramble or AAV-shRNA-Cx43 under NC or HFS diet conditions. (**D**–**G**) Quantification of microglial cell arborization in the DVC of mice treated with AAV-shRNA-Scramble or AAV-shRNA-Cx43 and fed with either NC or HFS diet. The total number of branches per microglial cell (**D**), total branch length (**E**), total arborization area (**F**), and average branch length (**G**) were quantified for the four experimental groups based on the skeleton of the arborization. (**H**) Illustration of confocal images processing during skeleton and Sholl analyses. (**I**) Sholl curves representing the number of intersections as a function of the distance from the soma (in µm) in mice treated with AAV-shRNA-Scramble or AAV-shRNA-Cx43, under NC or HFS diet conditions. Left graph compiles curves from the four groups. These curves illustrate microglia hyper-ramification observed in mice treated with AAV-shRNA-Cx43 compared to those that received AAV-shRNA-Scramble. (**J**–**L**) Microglial skeleton analysis quantified the total number of junctions ((**J**) points where two branches meet), total number of triple points ((**K**) points where three branches meet) and total number of quadruple points ((**L**) points where four branches meet). AAV-shRNA-Scramble_NC (n = 3, 55 cells); AAV-shRNA-Cx43_NC (n = 3, 55 cells); AAV-shRNA-Scramble_HFS (n = 3, 50 cells); and AAV-shRNA-Cx43_HFS (n = 3, 57 cells). Data are expressed as mean ± SEM. ** *p* < 0.01, and *** *p* < 0.001 indicates a significant difference from respective control. No significant difference was induced by the diet.

**Figure 5 cells-14-01694-f005:**
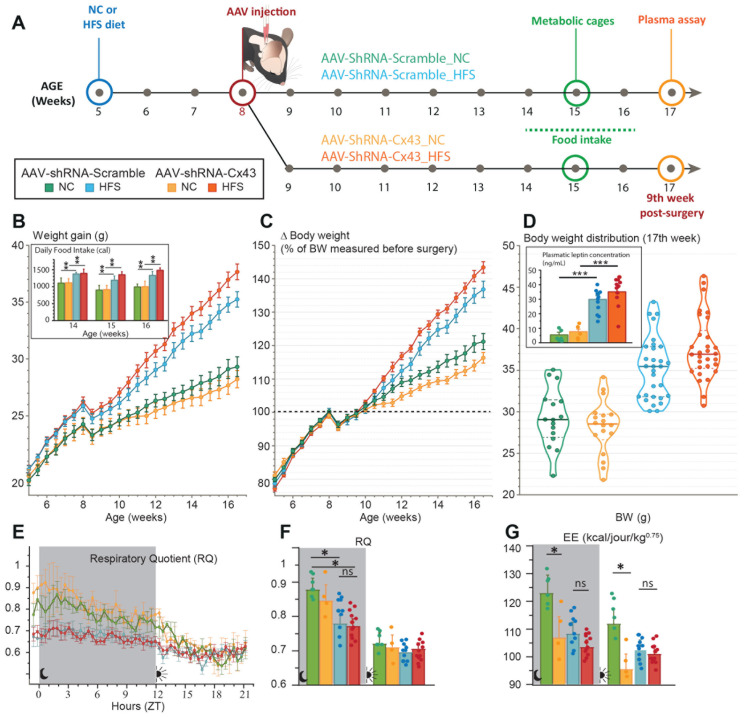
Weight gain, food intake and indirect calorimetry in Cx43 knockdown mice fed a HFS diet or maintained on a standard NC diet. (**A**) Experimental timeline. Two independent cohorts of mice were fed either a standard NC or HFS diet starting at 5 weeks of age (3 weeks before surgery). All mice received AAV-shRNA-Scramble or AAV-shRNA-Cx43 in the DVC at 8 weeks of age. All animals included in the metabolic test analysis underwent post-mortem examination to validate the incorporation of AAVs throughout the entire DVC. Animals excluded from post-mortem analysis, due to lack of complete AAV integration in the DVC, were not included in the results. (**B**–**D**) Weight gain (g) from the 3rd week before surgery and for 12 weeks (**B**), evolution of body weight expressed as a percentage relative to initial weight on the day of surgery (**C**) and to the final body weight 9 weeks post-injection (**D**). Data are expressed as mean ± SEM with two measurements per week. AAV-shRNA-Scramble_NC (n = 16); AAV-shRNA-Cx43_NC (n = 18); AAV-shRNA-Scramble_HFS (n = 30); AAV-shRNA-Cx43_HFS (n = 30). Daily food intake (in calories) (inset in (**B**)) was measured using the food residue weighing method. Food intake was not measured in animals participating in other metabolic tests. Data are expressed as mean ± SEM with two measurements per week. AAV-shRNA-Scramble-NC (n = 5); AAV-shRNA-Cx43-NC (n = 5); AAV-shRNA-Scramble_HFS (n = 11); AAV-shRNA-Cx43_HFS (n = 13). ** *p* <0.01 indicate significant difference between NC and HFS fed mice. Plasma leptin levels (inset in (**D**)) expressed in ng/mL in the different experimental groups. AAV-shRNA-Scramble_NC (n = 6); AAV-shRNA-Cx43_NC (n = 5); AAV-shRNA-Scramble_HFS (n = 12); AAV-shRNA-Cx43_HFS (n = 13). Data are expressed as mean ± SEM, *** *p* < 0.001 indicates significant difference from the respective NC-fed control groups. (**E**–**G**) Indirect calorimetry analysis. Respiratory quotient (RQ = VCO_2_/VO_2_; (**E**,**F**)), and energy expenditure (EE; (**G**)) were measured over a 24-h period in AAV-shRNA-Scramble or AAV-shRNA-Cx43 injected mice fed an NC or HFS diet. In panels F and G, the mean values were calculated for both the nocturnal and diurnal periods. AAV-shRNA-Scramble_NC (n = 6); AAV-shRNA-Cx43_NC (n = 4); AAV-shRNA-Scramble_HFS (n = 10); AAV-shRNA-Cx43_HFS (n = 12). Data are expressed as mean ± SEM. * *p* < 0.05 indicates a significant difference from AAV-shRNA-Scramble_NC mice. ns = not significant.

**Figure 6 cells-14-01694-f006:**
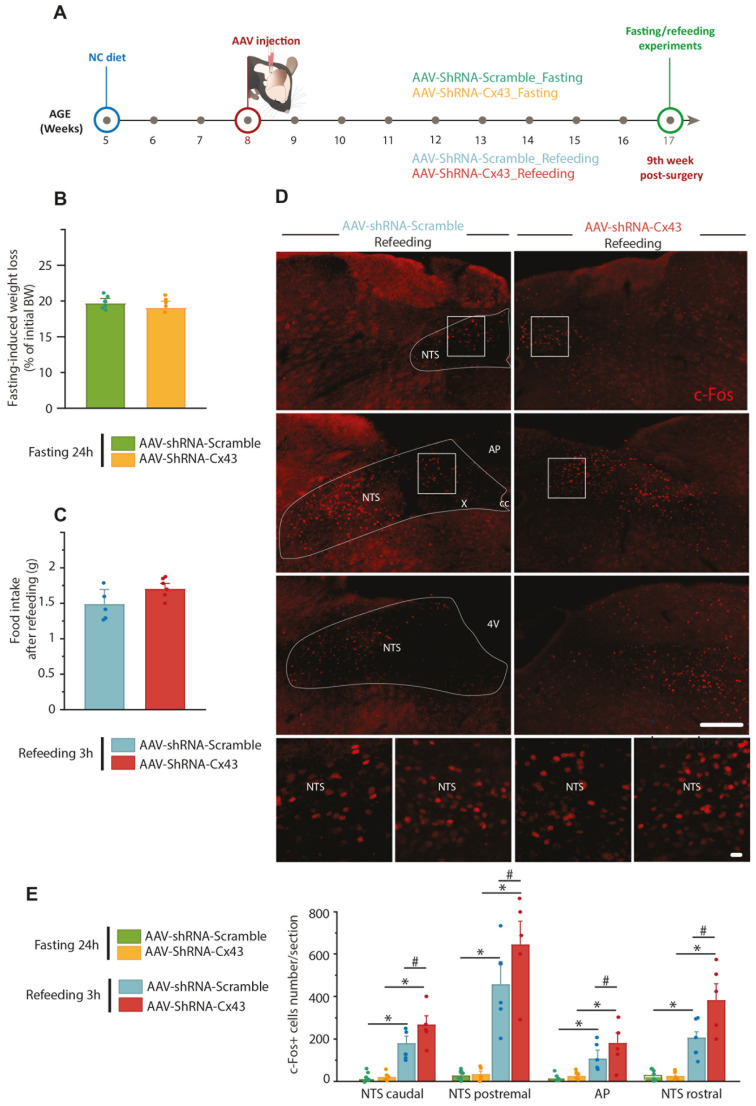
Metabolic responses to fasting and refeeding in Cx43 knockdown mice. (**A**) Experimental timeline. Mice used received AAV-shRNA-Scramble or AAV-shRNA-Cx43 in the DVC. Metabolic assays were performed 9 weeks after AAV injection. All animals included in the metabolic assays were subjected to post-mortem analysis to validate AAV incorporation into the entire DVC. (**B**,**C**) Change in body weight (% of initial body weight) induced after 21 h of fasting (**B**) and food intake per gram of body weight during the refeeding period (3 h) following fasting (**C**). (**D**) Representative c-Fos immunolabeling images (red) obtained by epifluorescence microscopy in the DVC of animals treated with AAV-shRNA-Scramble (**left**) or AAV-shRNA-Cx43 (**right**) fasted for 21 h then refed 3 h before sacrifice. High magnification images show enlargements of the caudal and postremal NTS, respectively. Squares indicate where high magnifications images originated. 4V: fourth ventricle, AP: area postrema; cc: central canal; X: motor nucleus of the vagus nerve; NTS: nucleus of the solitary tract. Scale bars: 200 µm (low magnifications images), 20 µm (high magnifications images). (**E**) Number of c-Fos+ cells in the caudal, postremal, rostral NTS and AP in mice perfused after a 24-h fast or a 21-h fast followed by a 3-h refeeding period. Data are expressed as mean ± SEM. AAV-shRNA-Scramble_fasting 24 h (n = 6); AAV-shRNA-Cx43_ fasting 24 h (n = 5); AAV-shRNA-Scramble_refeeding 3 h (n = 5); AAV-shRNA-Cx43_ refeeding 3 h (n = 5). * *p* < 0.05 indicates significant difference with respective control fasting group, # *p* < 0.05 indicates significant difference with AAV-shRNA-Scramble refed group.

**Figure 7 cells-14-01694-f007:**
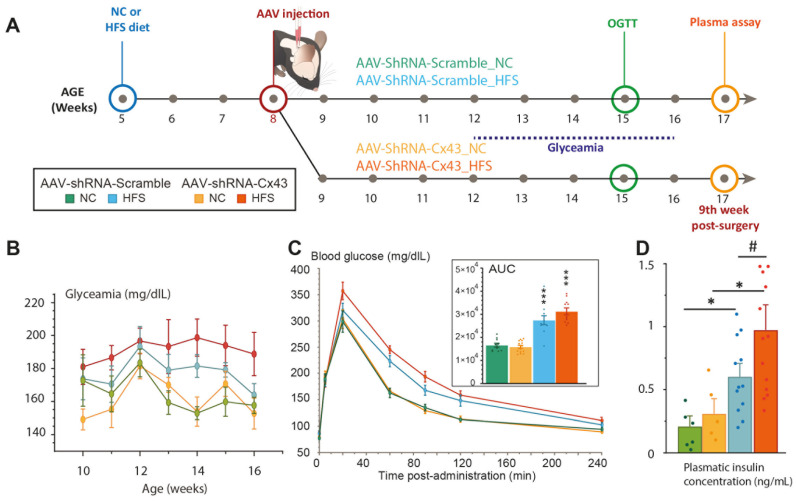
Glycemic status of Cx43 knockdown mice maintained on NC of HFS diet. (**A**) Experimental time line. (**B**) Evolution of blood glucose levels (mg/dL) between the 3rd and 9th week post-surgery. Food intake was not measured in animals during housing in metabolic cages. Data are expressed as mean ± SEM with two measurements per week. AAV-shRNA-Scramble_NC (n = 6); AAV-shRNA-Cx43_NC (n = 5); AAV-shRNA-Scramble_HFS (n = 12); AAV-shRNA-Cx43_HFS (n = 13). (**C**) Evolution of blood glucose (mg/dL) during the glucose tolerance test performed at 7th week post-surgery and area under the curve (AUC) of glucose levels measured during oral glucose tolerance tests. Blood glucose measurements are presented as a function of time after oral administration of glucose (1.5 g/kg). AAV-shRNA-Scramble_NC (n = 5); AAV-shRNA-Cx43_NC (n = 7); AAV-shRNA-Scramble_HFS (n = 11); AAV-shRNA-Cx43_HFS (n = 11). Data are expressed as mean ± SEM. *** *p* < 0.001 indicates significant difference with respective NC-fed control groups. (**D**) Plasma insulin levels expressed in ng/mL in the different experimental groups. AAV-shRNA-Scramble_NC (n = 6); AAV-shRNA-Cx43_NC (n = 5); AAV-shRNA-Scramble_HFS (n = 12); AAV-shRNA-Cx43_HFS (n = 13). Data are expressed as mean ± SEM. * *p* < 0.05, indicates significant difference with AAV-shRNA-Scramble fed NC control group. # *p* < 0.05, indicates significant difference with AAV-shRNA-Cx43 fed HFS control group.

**Figure 8 cells-14-01694-f008:**
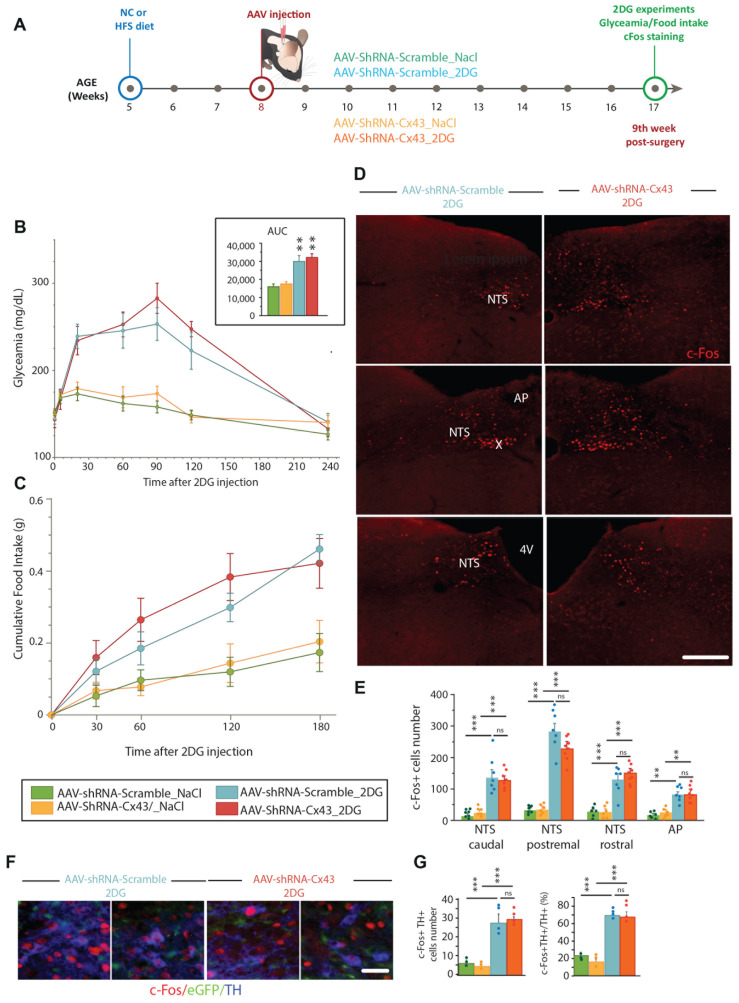
Metabolic responses to glucoprivation induced by 2DG administration in Cx43 knockdown mice. (**A**) Experimental time line. (**B**) Changes in blood glucose levels (mg/dL) and area under the curve (AUC) of glucose levels (inset) measured during the 2-DG-induced glucopenia test. Blood glucose measurements are shown as a function of time after 2DG administration (0.3 g/kg, 10 mL/kg) or NaCl (0.9%, 10 mL/kg). AAV-shRNA-Scramble_NaCl (n = 10); AAV-shRNA-Cx43_NaCl (n = 7); AAV-shRNA-Scramble_2DG (n =7); AAV-shRNA-Cx43_2DG (n = 6). Data are expressed as mean ± SEM. ** *p* < 0.01, indicates significant difference with AAV-shRNA-Scramble fed NC control group. (**C**) Cumulative food intake (grams) during the 3 h following administration of 2DG (0.3 g/kg, 10 mL/kg) or NaCl (0.9%, 10 mL/kg). Data are expressed as mean ± SEM. AAV-shRNA-Scramble_NaCl (n = 6); AAV-shRNA-Cx43_NaCl (n = 5); AAV-shRNA-Scramble_2DG (n =8); AAV-shRNA-Cx43_2DG (n =6). (**D**) Representative images of c-Fos immunostaining obtained by epifluorescence microscopy in animals treated with AAV-shRNA-Scramble (**left**) or AAV-shRNA-Cx43 (**right**) in the DVC, after 2DG injection. 4V: fourth ventricle, AP: area postrema; X: motor nucleus of the vagus nerve; NTS: nucleus of the solitary tract. Scale bar: 200 µm. (**E**) Number of c-Fos+ cells in caudal, postremal, rostral NTS and AP of mice perfused 2 h after injection of NaCl or 2DG (0.3 g/kg, 10 mL/kg). AAV-shRNA-Scramble_NaCl (n = 6); AAV-shRNA-Cx43_NaCl (n = 5); AAV-shRNA-Scramble_2DG (n = 8); AAV-shRNA-Cx43_2DG (n = 4). Data are expressed as mean ± SEM. ** *p* < 0.01, *** *p* < 0.001 indicates significant difference with NaCl-treated control group. No significant difference was induced by Cx43 deletion. (**F**). Images show enlargements of the caudal and postrema NTS respectively illustrating co-labelling of c-Fos (red), tyrosine hydroxylase immunoreactivity (TH+ neurons, blue) and EGFP fluorescence (AAV-transduced astrocytes, green). Scale bar: 20 µm. (**G**) Cell counting of c-Fos and tyrosine hydroxylase (TH) double-immunoreactive neurons and percentage of c-Fos + TH/TH neurons in NTS of mice perfused 2 h after injection of NaCl or 2DG (0.3 g/kg, 10 mL/kg). AAV-shRNA-Scramble_NaCl (n = 3); AAV-shRNA-Cx43_NaCl (n = 3); AAV-shRNA-Scramble_2DG (n = 4); AAV-shRNA-Cx43_2DG (n = 4). Data are expressed as mean ± SEM. *** *p* < 0.001 indicates significant difference with NaCl-treated control group. No significant difference was induced by Cx43 deletion. ns = not significant.

## Data Availability

The data presented in this study are available on request from the corresponding author due ethical reasons.

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
