# Peer review of "Glial Plasticity and Metabolic Stability After Knockdown of Astrocytic Cx43 in the Dorsal Vagal Complex"

_cells, 2025, doi:10.3390/cells14211694_

Round 1
Reviewer 1 Report
Comments and Suggestions for Authors
Major
- Time course and phenotyping window; Please provide a time-course quantification of Cx43 knockdown at 3, 5, and 9 weeks post-AAV (protein by Western; transcript by qPCR). This will define the onset and stabilization of KD. Several phenotypes may emerge only after a sufficiently sustained knockdown. Therefore, knowing when KD starts and plateaus is essential to justify that the week-9 readouts were collected at an appropriate physiological window.
- Durability and Late-Timepoint Phenotyping; Since AAV-shRNA often lasts for months in the CNS, it might help to add a later check (12–16 weeks) to see if the KD persists; if it does, looking at the key phenotypes at that time point
- Transduction efficiency and cell-type specificity; Although your bulk-tissue qPCR and IHC nicely demonstrate knockdown, because the AAV co-expresses EGFP it would strengthen the work to quantify transduction efficiency in the DVC by flow cytometry (percentage and MFI of EGFP⁺ astrocytes)and, ideally, to sort EGFP⁺ astrocytes and measure Cx43 by qPCR.
- Histology: n = 2 mice is insufficient; Some histology datasets use n = 2 mice. Even with many sections/cells, counts from the same mouse are pseudoreplicates and do not substitute for biological replicates. Please increase to at least 3–4 mice per group.
Minor
- Statistics notation; Please add connecting bars with asterisks to mark within-group tests in Figs. 3–4 and 6–7; readability suffers without explicit within-group bars.
- Figure 2 legend & A1 placement; The Figure 2 legend is overly dense and hurts readability. please simplify. Also, panels labeled as 2F–G are presented as Figure A1. this separation is unclear. Please either reintegrate them into Figure 2 or clearly justify and relabel their placement.
Author Response
Reviewer 1
Comments and Suggestions for Authors
Major
1. Time course and phenotyping window; Please provide a time-course quantification of Cx43 knockdown at 3, 5, and 9 weeks post-AAV (protein by Western; transcript by qPCR). This will define the onset and stabilization of KD. Several phenotypes may emerge only after a sufficiently sustained knockdown. Therefore, knowing when KD starts and plateaus is essential to justify that the week-9 readouts were collected at an appropriate physiological window.
This question is very important because it determines the timeframe during which deletion is effective. We evaluated AAV incorporation at week 3 and Cx43 deletion using quantitative PCR (qPCR) at week 4. At these time points, Cx43 deletion was already comparable to that observed after 9 weeks. These results were added to Ms in Figure S2. Therefore, Cx43 KD became efficient by the 3rd or 4th week after AAV injection and remained stable until week 9, enabling us to collect physiological data during this period.
2. Durability and Late-Timepoint Phenotyping; Since AAV-shRNA often lasts for months in the CNS, it might help to add a later check (12–16 weeks) to see if the KD persists; if it does, looking at the key phenotypes at that time point. In light of the results obtained in this study, it would be interesting to determine whether KD persists for longer than the 9 weeks examined here, and if so, what the mice's phenotype is at these later time points.
Nevertheless, we have demonstrated that a substantial deletion of astrocytic Cx43 over 9 weeks did not result in a notable metabolic phenotype, despite the glial cells of the structure responding strongly to this deletion. We believe that the discrepancy between impaired glial communication and the maintenance of metabolic functions deserves to be described, although it cannot be ruled out that the metabolic phenotype changed after 12–16 weeks.
3. Transduction efficiency and cell-type specificity; Although your bulk-tissue qPCR and IHC nicely demonstrate knockdown, because the AAV co-expresses EGFP it would strengthen the work to quantify transduction efficiency in the DVC by flow cytometry (percentage and MFI of EGFP⁺ astrocytes) and, ideally, to sort EGFP⁺ astrocytes and measure Cx43 by qPCR.
We thank the referee for highlighting that we clearly demonstrated the efficiency and specificity of the deletion. Their suggestion is relevant and would undoubtedly have complemented our description of the deletion usefully. Unfortunately, conducting this type of experiment with new animals requires authorization from local and national ethics committees, as well as a timeframe for the experiment that is compatible with the deadline for the revision.
4. Histology: n = 2 mice is insufficient; Some histology datasets use n = 2 mice. Even with many sections/cells, counts from the same mouse are pseudoreplicates and do not substitute for biological replicates. Please increase to at least 3–4 mice per group.
The number of mice has been increased to reach 3-4 mice per group.
Minor
1. Statistics notation; Please add connecting bars with asterisks to mark within-group tests in Figs. 3–4 and 6–7; readability suffers without explicit within-group bars
Connecting bars have been added.
2. Figure 2 legend & A1 placement; The Figure 2 legend is overly dense and hurts readability. please simplify. Also, panels labeled as 2F–G are presented as Figure A1. this separation is unclear. Please either reintegrate them into Figure 2 or clearly justify and relabel their placement.
These points have been corrected, and the Figure 2 legend simplified.
Reviewer 2 Report
Comments and Suggestions for Authors
Authors have focused on the dorsal vagal complex (DVC) region to investigate the role of glial cells, especially astrocytes and microglia, in regulating metabolic regulation. They have used an AAV-ShRNA approach to eliminate CX43 from astrocytes residing in the DVC region. They have then performed several animal studies to analyze the response of glial cells in normal feeding conditions versus high-fat diet/high-sugar conditions. The Materials and Methods section is well-written; however, I have some major concerns:
- The Introduction, Results, and Discussion sections are unnecessarily long, which makes it difficult to follow.
- The conclusions drawn from the data are a bit overestimated.
- ALL figures (Fig. 1, Fig. 2, Fig. 3, Fig. 4, Fig. 6, Fig. 8) are missing the DAPI channel.
- Some figures are missing Y-axis labels (e.g. Fig. 2 I, J).
- Please show individual data points on bar graphs.
- It is very difficult to conclude anything from the c-Fos staining, as the DAPI channel is missing.
- Also, c-Fos staining alone without other cell type markers is missing.
- Overall, the study does not provide enough evidence to support the title of the article and the claims that the authors have made.
Author Response
Reviewer 2
Comments and Suggestions for Authors
Authors have focused on the dorsal vagal complex (DVC) region to investigate the role of glial cells, especially astrocytes and microglia, in regulating metabolic regulation. They have used an AAV-ShRNA approach to eliminate CX43 from astrocytes residing in the DVC region. They have then performed several animal studies to analyze the response of glial cells in normal feeding conditions versus high-fat diet/high-sugar conditions. The Materials and Methods section is well-written; however, I have some major concerns:
1. The Introduction, Results, and Discussion sections are unnecessarily long, which makes it difficult to follow. The introduction and discussion have been extensively revised and shortened to make the message clearer.
2. The conclusions drawn from the data are a bit overestimated.
As suggested, this part has been modified as follows:
5.Conclusions
Overall, our results show that disruption of astrocytic communication in the DVC via Cx43 deletion is accompanied by a strong astrocytic and microglial reaction. Astrocyte densification and microglial hyper-ramification could constitute an adaptive system capable of preserving vital functions operating at the brainstem level, such as metabolic and autonomic regulations, at least in the first few weeks. Although still hypothetical, this possibility warrants testing under longer and/or other conditions of autonomic dysfunction, such as more severe metabolic overload or prolonged hypoxia conditions [8].
3. ALL figures (Fig. 1, Fig. 2, Fig. 3, Fig. 4, Fig. 6, Fig. 8) are missing the DAPI channel. DAPI was not systematically added to our immunochemistry experiments because it seemed not really necessary given the information researched and/or the blue channel was used for other labeling.
4. Some figures are missing Y-axis labels (e.g. Fig. 2 I, J). The Y-axis labels were initially placed above figures 2iI, J, but we moved them to the side of the Y-axis.
5. Please show individual data points on bar graphs. As suggested, this information has been added on Figures 3,4, 5, 6, 7 and 8.
6. It is very difficult to conclude anything from the c-Fos staining, as the DAPI channel is missing. This comment is surprising. C-Fos labelling would be useless without DAPI labelling. Does this mean that all studies in which c-Fos labelling was revealed by DAB without nuclear counterstaining are invalid? c-Fos labelling has frequently been used by us and others, and the results shown here are typical of nuclear labelling. Figure 8F provides evidence of this with the TH/c-Fos labelling. DAPI was not consistently added to our immunochemistry investigations either because it was unnecessary, or because the blue channel was used for other staining.
7. Also, c-Fos staining alone without other cell type markers is missing. Which phenotypic markers would have been relevant to use here? Our objective was not to determine which neuronal populations were activated, as this has been done extensively elsewhere. Rather, our objective was to determine whether the overall cellular (neuronal) activation known to occur in these experimental situations was altered. We performed phenotypic labelling of TH neurons when we felt it was important, as these neurons are known to play a crucial role in the response to cellular glucopenia.
8. Overall, the study does not provide enough evidence to support the title of the article and the claims that the authors have made. We disagree with this comment. The title we chose is factual: it refers to the glial modification and the lack of a significant metabolic phenotype observed following the deletion of Cx43 in DVC astrocytes. We do not believe that the title overinterprets any of the results. We would like to keep this title because we believe it accurately summarizes the results and provides potential readers with the right information.
Reviewer 3 Report
Comments and Suggestions for Authors
There were some minor concerns in the present manuscript.
For the citation in text, the numbers were used in the journal "Cells". Therefore, all "Author's name et al., year" after the reference numbers should be deleted.
In Introduction, abbreviation "HC" was not explained.
In 3. Results, "3.1.1.~3.1.5." should be corrected as "3.1~3.5.", and "3.2. Figures, Tables and Schemes" should be deleted.
Line 430-431, "(Figure 3E-G)" should be corrected as "(Figure 3E)". And add "(Figure 3G)" after "or a HFS diet" in line 432.
Line 487, was the referenced figure number (Figure 4D-L) correct?
Figure 3F and the text, HFS-fed AAV-shRNA-Cx43 mice do not display the high astrocytic size observed in AAV-shRNA-Cx43 fed a NC diet. Why? Knockdown of astrocytic Cx43 significantly increased the cell size. And, NC and HFS diets did not affect the cell size in AAV-shRNA-Scramble. Therefore, why did HFS diet inhibit the increase of cell size in AAV-shRNA-Cx43?
In Figure 2B-G, what did the arrowheads show?
Line 655, correct "Figure A1".
Author Response
For the citation in text, the numbers were used in the journal "Cells". Therefore, all "Author's name et al., year" after the reference numbers should be deleted. This point has been corrected.
In Introduction, abbreviation "HC" was not explained. Done.
In 3. Results, "3.1.1.~3.1.5." should be corrected as "3.1~3.5.", and "3.2. Figures, Tables and Schemes" should be deleted. Done.
Line 430-431, "(Figure 3E-G)" should be corrected as "(Figure 3E)". And add "(Figure 3G)" after "or a HFS diet" in line 432. Done.
Line 487, was the referenced figure number (Figure 4D-L) correct? Yes correct.
Figure 3F and the text, HFS-fed AAV-shRNA-Cx43 mice do not display the high astrocytic size observed in AAV-shRNA-Cx43 fed a NC diet. Why? Knockdown of astrocytic Cx43 significantly increased the cell size. And, NC and HFS diets did not affect the cell size in AAV-shRNA-Scramble. Therefore, why did HFS diet inhibit the increase of cell size in AAV-shRNA-Cx43? That's an interesting question, but unfortunately, we don't have an explanation at this stage. We can only assume that the astroglial plasticity observed in responses to Cx43 KD is influenced by the environment provided by the diet. Since HFD alone was reported to induced astrocytes remodeling in the DVC (MacDonald et al., Glia 2020), this hypothesis seems conceivable.
In Figure 2B-G, what did the arrowheads show? Precisions have been added.
Line 655, correct "Figure A1". This point has been corrected.
Round 2
Reviewer 1 Report
Comments and Suggestions for Authors
The authors have addressed the majority of the comments clearly and constructively. The revised manuscript shows marked improvement in clarity, methodological transparency, and overall figure presentation. The authors’ responses demonstrate solid scientific reasoning, and the added revisions significantly strengthen the manuscript. While a few suggestions were not implemented due to practical limitations, the justifications are reasonable and do not affect the main conclusions. Overall, the authors have effectively incorporated the reviewers’ comments and improved the quality of the manuscript.
Author Response
We are very grateful to the reviewer for these positive comments and for the quality of their remarks during the review process.
Reviewer 2 Report
Comments and Suggestions for Authors
Thank you for the response. I appreciate that the authors have done a good job of restructuring the manuscript.
1. I still believe that DAPI channel is essential these days to show the individual cell types. Although authors have used DAPI channel for other staining, they could have used four channels to represent all three staining plus DAPI channel. If authors have published this type of data in the past and others have done so, that does not mean it is the correct way.
2. Regarding c-Fos, I agree it is nuclear marker and perhaps you don't need DAPI channel to show the nuclie as well. However, it is good to know what proportion of the DAPI positive nuclie show c-Fos staining? Also, this has to be per area and not per section.
3. Authors have shown co-localisation of TH+ Neurons with c-Fos. However, if you look close there are other non TH+cells showing expression of c-Fos. Any comment on this?
4. Authors have shown quantification of c-Fos+ cells throughout the manuscript. Authors stated in the response that "our objective was to determine whether the overall cellular (neuronal) activation known to occur in these experimental situations was altered". Don't you think the quantification should show TH+ and c-Fos+ neuronal cells?
Author Response
Thank you for the response. I appreciate that the authors have done a good job of restructuring the manuscript.
- I still believe that DAPI channel is essential these days to show the individual cell types. Although authors have used DAPI channel for other staining, they could have used four channels to represent all three staining plus DAPI channel. If authors have published this type of data in the past and others have done so, that does not mean it is the correct way.
We agree with the referee that today, perhaps more than yesterday, DAPI staining may be useful in certain experimental conditions to identify each cell. We could have used four channels, but we didn't think it would add value to the present manuscript. We note that the referee considers our method to be incorrect. Nevertheless, we invite him (her) to consult publications, which, although recent (2024-2025), did not perform DAPI labeling in addition to c-Fos labeling in the NTS.
- J Neurosci 2024 Aug 21;44(34):e0084242024. doi: 10.1523/JNEUROSCI.0084-24.2024.
- Front Neurosci 2024 Apr 29:18:1401530. doi: 10.3389/fnins.2024.1401530.
- Sci Rep 2024 Mar 29;14(1):7473. doi: 10.1038/s41598-024-58075-x.
- Am J Physiol Regul Integr Comp Physiol 2024 May 1;326(5):R383-R400. doi: 10.1152/ajpregu.00238.2022.
- Pflugers Arch 2024 Jul;476(7):1087-1107. doi: 10.1007/s00424-024-02957-6.
- Neurobiol Stress 2025 Apr 1:36:100723. doi: 10.1016/j.ynstr.2025.100723.
- Regarding c-Fos, I agree it is nuclear marker and perhaps you don't need DAPI channel to show the nuclie as well. However, it is good to know what proportion of the DAPI positive nuclie show c-Fos staining? Also, this has to be per area and not per section.
We are pleased that the referee agrees with us that, given the highly characteristic nature of c-Fos staining, it is not essential to show the nucleus with DAPI. However, we find very surprising the referee’s comment suggesting using DAPI staining to determine the proportion of nuclei that are positive for DAPI and c-Fos. DAPI staining will highlight neurons but also astrocytes, microglial cells, and oligodendrocytes. We really do not see the rationale for determining the percentage of c-Fos+ neurons in relation to all the cells in the structure.
- Authors have shown co-localisation of TH+ Neurons with c-Fos. However, if you look close there are other non TH+cells showing expression of c-Fos. Any comment on this?
As observed by the reviewer, some non-TH cells are positive for c-Fos. This point has been widely described in the literature. It has been shown that while a large majority of TH neurons in the NTS are activated in response to glucoprivic stimuli (Briski and Marshall, Exp Brain Res 2000 Aug;133(4):547-51. doi: 10.1007/s002210000448), these neurons represent only a fraction of c-Fos+ neurons (8.3% according to Moriyama et al., J Reprod Dev 2003 Apr;49(2):151-7. doi: 10.1262/jrd.49.151). We did not include this information in the discussion because we did not consider it a priority. We can do so if the referee considers that this point deserves to be added.
- Authors have shown quantification of c-Fos+ cells throughout the manuscript. Authors stated in the response that "our objective was to determine whether the overall cellular (neuronal) activation known to occur in these experimental situations was altered". Don't you think the quantification should show TH+ and c-Fos+ neuronal cells?
We fully agree with the reviewer. Consequently, we have added quantification of the number of c-Fos+ TH neurons as well as the percentage of c-Fos+ TH/TH neurons (see Figure 8G).
Round 3
Reviewer 2 Report
Comments and Suggestions for Authors
No further comments.